# Vocal and locomotor coordination develops in association with the autonomic nervous system

Morgan L Gustison[1†], Jeremy I Borjon[1,2], Daniel Y Takahashi[1,2]*, Asif A Ghazanfar[1,2,3]*

[1]Princeton Neuroscience Institute, Princeton University, Princeton, United States; [2]Department of Psychology, Princeton University, Princeton, United States; [3]Department of Ecology and Evolutionary Biology, Princeton University, Princeton, United States

**Abstract** In adult animals, movement and vocalizations are coordinated, sometimes facilitating, and at other times inhibiting, each other. What is missing is how these different domains of motor control become coordinated over the course of development. We investigated how postural-locomotor behaviors may influence vocal development, and the role played by physiological arousal during their interactions. Using infant marmoset monkeys, we densely sampled vocal, postural and locomotor behaviors and estimated arousal fluctuations from electrocardiographic measures of heart rate. We found that vocalizations matured sooner than postural and locomotor skills, and that vocal-locomotor coordination improved with age and during elevated arousal levels. These results suggest that postural-locomotor maturity is not required for vocal development to occur, and that infants gradually improve coordination between vocalizations and body movement through a process that may be facilitated by arousal level changes.
DOI: https://doi.org/10.7554/eLife.41853.001

*For correspondence:
takahashiyd@gmail.com (DYT);
asifg@princeton.edu (AAG)

Present address: †Department of Integrative Biology, University of Texas at Austin, Austin, United States

Competing interests: The authors declare that no competing interests exist.

## Introduction

Vocal development is typically thought of as the adaptive coordination of the vocal apparatus (i.e., the lungs, larynx and the mouth) and the associated muscles and neural systems that influence its activity. In adult animals, however, vocal behavior does not occur in isolation of the operations of other motor systems. Studies investigating the real-time coordination between vocal and locomotor outputs show that some vocalizations can be produced concurrently with locomotor activity and/or postural changes, while others cannot (*Suthers et al., 1972*; *Blumberg, 1992*; *Fusani et al., 1996*; *Williams, 2001*; *Wong and Waters, 2001*; *Holderied and von Helversen, 2003*; *Branchi et al., 2004*; *Cooper and Goller, 2004*; *Berg et al., 2013*; *Dalziell et al., 2013*; *Hoepfner and Goller, 2013*; *Ota et al., 2015*; *Alves et al., 2016*; *Laplagne and Elías Costa, 2016*; *Ullrich et al., 2016*). For example, the 'A', 'B' and 'D' song types of the lyrebird (*Menura novaehollandiae*) co-occur with courtship dance wing flaps, while the 'C' type occurs when wings are still (*Dalziell et al., 2013*). In rats (*Rattus norvegicus*), 50 kHz ultrasonic vocalizations are produced during locomotor activity, while 20 kHz vocalizations occur when rats are immobile (*Laplagne and Elías Costa, 2016*). Similarly, bats in flight coordinate the production of echolocation sounds with particular phases of their wing beats (*Suthers et al., 1972*; *Wong and Waters, 2001*; *Holderied and von Helversen, 2003*). Thus, the adaptive coordination achieved during vocal development must also include other developing motor systems as important factors, notably those related to posture and locomotion. How this 'vocal-locomotor' coordination is accomplished over the course of development is not well understood.

In humans, current theoretical frameworks posit that the different motor systems are interactive over the course of development; one system can influence another in ways that change as a function of time (*Thelen, 1991*; *Adolph, 2008*; *Iverson, 2010*; *Adolph and Robinson, 2015*; *Libertus and Hauf, 2017*). In support of this, there is evidence that locomotor skills at one time point predict speech ability months or even years later (*LeBarton and Iverson, 2013*; *Walle and Campos, 2014*; *Wang et al., 2014*; *He et al., 2015*; *LeBarton and Iverson, 2016*; *Libertus and Violi, 2016*; *Walle, 2016*; *Garrido et al., 2017*; *Libertus and Hauf, 2017*; *Salavati et al., 2017*; *West et al., 2019*). However, only a handful of empirical studies investigated human infant vocal-locomotor coordination (*Ejiri and Masataka, 2001*; *Fagan and Iverson, 2007*; *Abney et al., 2014*; *Berger et al., 2017*). These studies showed that infant production of pre-linguistic vocalization is highly sensitive to body movement. For example, infants who are beginning to transverse their environment (i.e., crawling) are less likely to vocalize during locomotion than while they are sitting (*Ejiri and Masataka, 2001*; *Fagan and Iverson, 2007*; *Abney et al., 2014*; *Berger et al., 2017*). What is missing is an understanding of *how* vocal and locomotor outputs are coordinated in real-time in infants and how this coordination may change over the course of development.

Key to understanding these developmental dynamics is to also identify physiological conditions that may promote (or potentially inhibit) coordination between different motor outputs. One candidate is the state of arousal, a product of the autonomic nervous system and relevant for a range of behaviors (*Pfaff, 2006*). An animal would be said to exhibit a high arousal state if it is more alert to sensory stimuli, more motorically active and more reactive (*Pfaff, 2006*). The role of arousal is essentially to allocate metabolic energy (i.e., to prepare the body for action). As it relates to vocal production, arousal modulates respiration, which in turn provides the power for vocal output. Humans, for example, exhibit an increase in arousal–as measured by heart rate–prior to speaking (*Lynch et al., 1980*; *Linden, 1987*). In developing individuals, variable and spontaneous behaviors are ubiquitous, providing the scaffolding for more complex and organized behaviors later in life (*Blumberg et al., 2013*). These early behaviors, including vocal output and other bodily movements, primarily reflect the interplay between the infants' arousal states, sensorimotor coordination and biomechanical conditions (*Robinson et al., 2000*). Thus, investigating the relationship between arousal fluctuations and the development of vocal and locomotor behaviors may prove to be illuminating.

Using marmoset monkeys as a model, here we investigate the relationship between vocal and locomotor systems and arousal levels during postnatal development. In the vocal domain, infant marmosets spontaneously produce sequences of immature and mature vocalizations, and these are linked to real-time changes in arousal levels (*Zhang and Ghazanfar, 2016*). Over the course of approximately two months, infants exhibit changes in the acoustic properties of their vocalizations that reflect a transition from producing mostly immature-sounding contact calls (e.g., cries) to mature-sounding contact calls (e.g., phees) (*Takahashi et al., 2015*; *Zhang and Ghazanfar, 2016*; *Teramoto et al., 2017*; *Zhang and Ghazanfar, 2018*). As in humans (*Goldstein and Ja, 2008*), this transition is facilitated by, and dependent upon, social reinforcement from parents (*Takahashi et al., 2015*; *Gultekin and Hage, 2017*; *Takahashi et al., 2017*; *Gultekin and Hage, 2018*). Moreover, these parallels with human prelinguistic development occur in the same life history stage (early infancy) (*Ghazanfar and Liao, 2018*). In the postural and locomotor domains, marmoset monkey development transitions from immature to mature forms in a pattern that is also similar to human development (e.g., righting reflex before sitting, and crawling before walking) (*Wang et al., 2014*; *Braun et al., 2015*; *Schultz-Darken et al., 2016*).

We address three fundamental questions: (1) Does one motor system – vocal or postural-locomotor – mature first or do they follow an overlapping trajectory? (2) How are vocal-postural-locomotor systems coordinated and do these coordination dynamics shift across development? (3) How do real-time fluctuations in arousal relate to vocal production, locomotion and their coordination?

## Results

During their first two months of postnatal life, we measured infant marmoset behavior in a controlled context for 10 minutes approximately every 2 days. In each session, individuals were placed in a testing box in an experiment room that was outside visual and auditory range of their family groups. This brief 'isolation' context is a standard testing paradigm used to elicit vocalizations (*Takahashi et al., 2015*; *Zhang and Ghazanfar, 2016*) and to study the postures and locomotion of

marmoset infants (*Wang et al., 2014*; *Braun et al., 2015*). The subjects were seven marmosets (three females) from three different parental pairs (two sets of twins, one set of triplets). We recorded the behaviors of each subject across ~30 sessions (28–33 per subject for a total of 220 sessions).

For *vocal* behaviors, we focused on two types of contact calls – cries and phees (*Figure 1A*). As described previously (*Takahashi et al., 2015*), cries are immature contact calls that have a short duration and noisy spectral properties (i.e., high Wiener entropy); phees are mature-sounding contact calls that have a longer duration and tonal spectral properties (i.e., low Wiener entropy). Cries transform into phees over the course of development (*Takahashi et al., 2015*; *Zhang and Ghazanfar, 2016*; *Takahashi et al., 2017*).

We measured five types of *postural* behaviors – righting reflex, head raising, forelimb support, hindlimb support, and hanging (*Wang et al., 2014*; *Braun et al., 2015*) (*Figure 1B*). The righting reflex is when infants re-establish their body orientation so that their hands and feet are on the ground; head raising is when infants lift their head off the ground and look forward or up; forelimb support is when infants sit on the ground with their hands touching the ground; hindlimb support is when infants sit on the ground with their hands off the ground; hanging is when infants grasp elements in their environment (e.g., bars of the testing box) so that their hands and feet do not touch the ground.

We measured five types of *locomotor* behaviors – crawling, digging, jumping, climbing, and walking (*Wang et al., 2014*; *Braun et al., 2015*) (*Figure 1C*). Crawling is when infants move forward on the ground with their stomach touching the ground; digging is when infants move their hands back and forth across the ground; jumping is when infants push themselves off the ground or cage to move from one location to another; climbing is when infants traverse across the cage; walking is when infants traverse across the ground in a standing orientation.

Finally, for all seven infants, we concurrently measured arousal levels by acquiring heart rates during the sessions using non-invasive surface electrocardiography (*Borjon et al., 2016*; *Zhang and Ghazanfar, 2016*).

## The vocal system matures before postural and locomotor systems

By measuring both vocal and postural-locomotor behaviors longitudinally in developing marmosets, we determined how these motor systems changed relative to one another. We first classified each behavior as immature or mature by measuring how their use shifted across development. In the vocal domain, the proportion of time producing cries decreased across development, while the proportion of time producing phees increased (*Figure 2A*; *Table 1*; Appendix 1.1) (*Takahashi et al., 2015*; *Zhang and Ghazanfar, 2016*). As indicated by physiology and biomechanics (*Takahashi et al., 2015*; *Zhang and Ghazanfar, 2016*; *Zhang and Ghazanfar, 2018*), cries were categorized as an immature contact call, and phees were categorized as a mature version of the contact

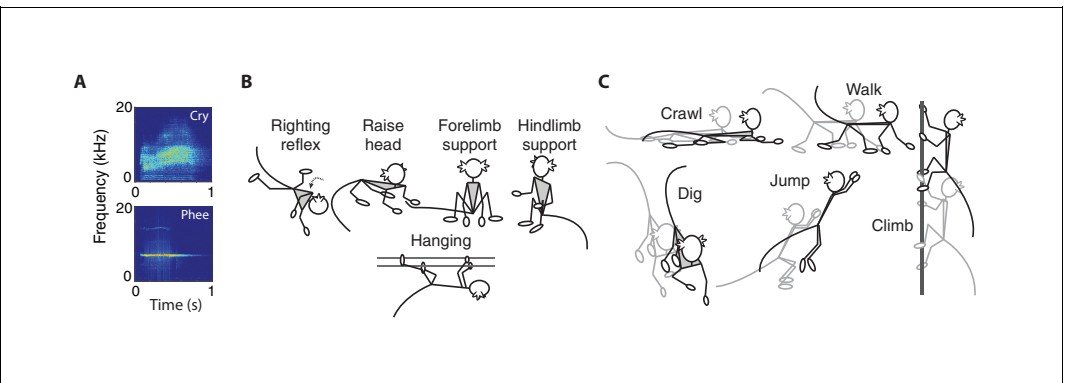

**Figure 1.** Vocal, postural and locomotor behaviors in developing marmosets. (**A**) Spectrograms of immature (cries) and mature (phees) contact calls produced by infant marmosets in an isolated social context. (**B**) Cartoons representing the five types of posture behaviors. (**C**) Cartoons representing the five types of locomotor behaviors.
DOI: https://doi.org/10.7554/eLife.41853.002

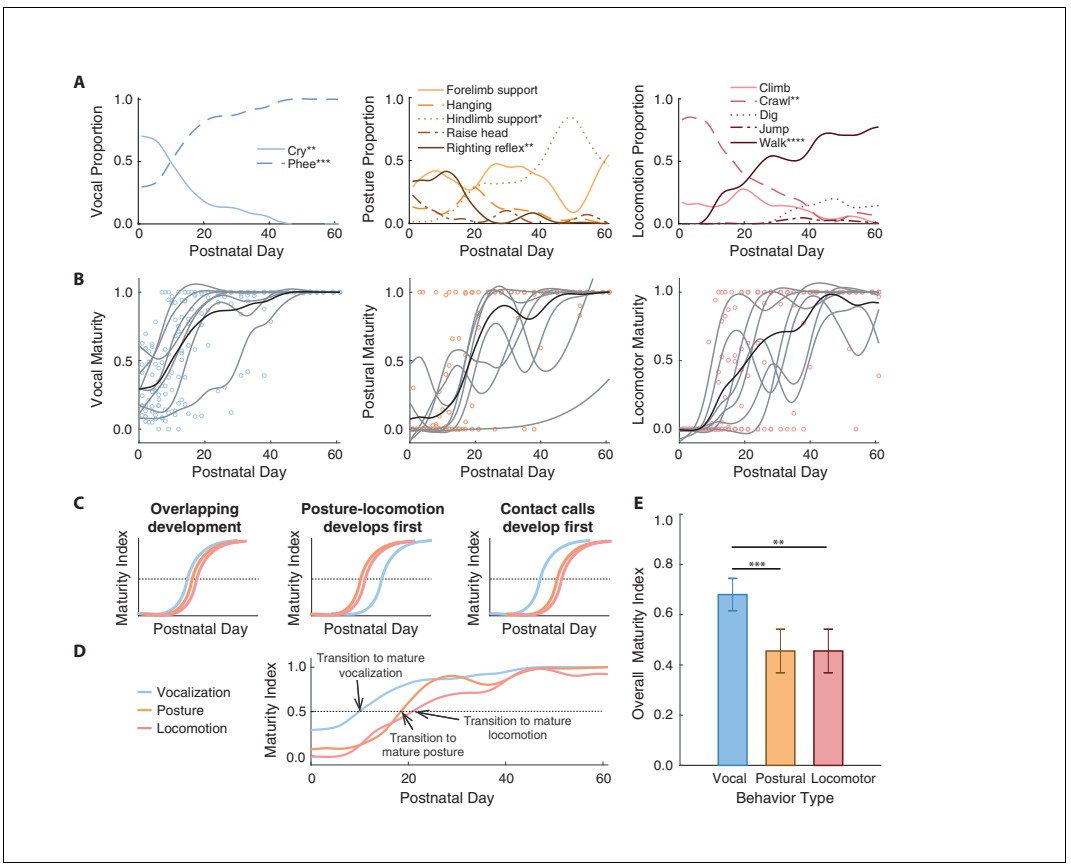

**Figure 2.** Contact calls mature before postural and locomotor behaviors. (A) Changes in the proportion of time spent in specific vocal, postural, and locomotor behaviors across development. Behaviors that increase or decrease are denoted with asterisks to represent significance values. (B) Developmental trajectories of contact call maturation (i.e., mature phee calls relative to immature cries), postural maturation (i.e., hindlimb support relative to righting reflex), and locomotor maturation (i.e., walking relative to crawling). Points represent session values, gray curves represent the cubic spline fits for individual marmosets, and black curves represent the population cubic spline fits. (C) Illustration of hypothesized sequences between contact call and postural/locomotor developmental trajectories and (D) an overview of the observed developmental trajectories. (E) Comparison of the different developmental time courses. Bars and whiskers represent mean ±2 SE. Significance values are represented by '*' (p<0.05), '**' (p<0.01), '***' (p<0.001) and '****' (p<0.0001).

DOI: https://doi.org/10.7554/eLife.41853.003

call. The proportion of postural time engaged in righting reflexes decreased across development, while the proportion of postural time engaged in hindlimb support increased (*Figure 2A*; *Table 1*; Appendix 1.2). Thus, righting reflexes were categorized as an immature posture, and hindlimb support was categorized as a mature posture. The proportion of locomotor time engaged in crawling decreased across development, while the proportion of locomotor time engaged in walking increased (*Figure 2A*; *Table 1*; Appendix 1.3). Thus, crawling was categorized as an immature locomotor behavior, while walking was categorized as mature. These postural-locomotor classifications for developing marmosets are consistent with previous findings (*Wang et al., 2014*).

We used the immature-mature classifications to summarize developmental changes using a 'maturity index' (see Materials and methods)–a value that represents the proportion of mature behaviors observed relative to all immature and mature behaviors across postnatal days. Values below 0.5 indicate that immature behavior is more common, and values above 0.5 indicate that mature behavior is more common. The population-level developmental trajectories of all three motor systems (vocal = 192 observation days, postural = 201 observation days, locomotor = 191 observation days) followed s-shape patterns that ended in maturity indices around one, which represents adult-like motor outputs (*Figure 2B*).

**Table 1.** Results of linear mixed models (LMMs) to test whether proportion of time spent in vocal-postural-locomotor behaviors changes with postnatal day.

For each model, the proportion of vocal, postural, or locomotor time (per postnatal day) spent engaged in a behavior is the dependent variable, postnatal day is the fixed effect, and infant identity is the random effect. For each behavior category, a Bonferroni-Holmes correction was applied to adjust p-values.

| Behavior | β (SE) | T value | Adjusted P | Classification |
|---|---|---|---|---|
| *Vocal behaviors* (n = 192 observation days) | | | | |
| Cry | −0.0128 (0.0019) | 6.88 | 0.0011 | immature |
| Phee | 0.0129 (0.0017) | 7.43 | 0.0004 | mature |
| *Postural behaviors* (n = 201 observation days) | | | | |
| Forelimb support | −0.0010 (0.0020) | 0.51 | >0.05 | NA |
| Hanging | −0.0018 (0.0011) | 1.67 | >0.05 | NA |
| Hindlimb support | 0.0130 (0.0034) | 3.78 | 0.0373 | mature |
| Raising head | −0.0021 (0.0009) | 2.32 | >0.05 | NA |
| Righting reflex | −0.0081 (0.0015) | 5.54 | 0.0020 | immature |
| *Locomotor behaviors* (n = 191 observation days) | | | | |
| Climbing | −0.0023 (0.0011) | 2.10 | >0.05 | NA |
| Crawling | −0.0160 (0.0019) | 8.35 | 0.0019 | immature |
| Digging | 0.0033 (0.0011) | 3.03 | >0.05 | NA |
| Jumping | 0.0005 (0.0002) | 2.37 | >0.05 | NA |
| Walking | 0.0147 (0.0012) | 11.95 | <0.0001 | mature |

DOI: https://doi.org/10.7554/eLife.41853.004

We tested whether the developmental time courses of the vocal system and postural-locomotor systems overlap (*Figure 2C*). The null hypothesis is that vocal and postural-locomotor systems transition from immature to mature forms around the same time. This would suggest that the development of the different motor systems reflect a global process of neural or physiological maturation (*Gesell, 1929*; *McGraw, 1943*). The alternative hypothesis is that the postural-locomotor system develops either before or after the vocal system. If postural-locomotor behaviors were to mature first, it would suggest that a developed post-cranial body is a prerequisite for producing mature contact calls.

Supporting the 'vocal behavior develops first' hypothesis, contact calls transitioned to a more mature form around 10 postnatal days, while postural and locomotor behavior transitioned around postnatal days 19 and 21, respectively (*Figure 2D*). A generalized linear mixed model (GLMM) showed that the relationship between maturity indices and postnatal day fits a logistic regression, with maturity indices increasing with age (n = 519 observation days, β ±SE = 0.21±0.03, z = 6.33, p<0.0001; Appendix 1.4). The same model showed that the vocal maturity indices were larger than the maturity indices of postural (β ±SE = −2.03±0.58, z = 3.48, p=0.0005; Appendix 1.4) and loco-motor behavior (β ±SE = −2.86±0.98, z = 2.93, p=0.0034; Appendix 1.4) (*Figure 2E*). In other words, vocal development occurred earlier than postural and locomotor development. However, simply because vocal-postural-locomotor systems follow different trajectories does not mean that they do not interact in real-time. The question of whether vocal-locomotor coordination changes over the course of development is addressed next.

## Mature contact call production and locomotor activity become increasingly coordinated during development

Infant marmosets require considerable muscular effort to produce mature contact calls (phees) (*Takahashi et al., 2015*; *Zhang and Ghazanfar, 2016*; *Zhang and Ghazanfar, 2018*), and adult marmosets tend to produce mature contact calls during periods of reduced locomotor activity (*Borjon et al., 2016*). Therefore, our overarching hypothesis was that locomotor activity inhibits

mature contact call production. Using linear mixed effect models (LMMs), we found initial support for this hypothesis when examining the relationships between vocal acoustic parameters, locomotor activity and postnatal day (*Figure 3A,B*). Call duration was negatively associated with locomotor activity (n = 9609 calls, β ±SE = −0.56±0.13, $t$ = 4.25, Bonferroni-Holm adjusted p=0.0001; Appendix 1.5) while Wiener entropy (higher entropy means noisier) was positively associated with locomotor activity (n = 9606 calls, β ±SE = 0.93±0.12, $t$ = 7.68, Bonferroni-Holm adjusted p<0.0001; Appendix 1.6). In other words, on average, infant marmosets produced short, noisier calls when locomotor activity was high, and longer, more tonal adult-like calls when locomotor activity was reduced.

Next, we tested three hypotheses about the developmental dynamics of vocal-locomotor coordination. We know that by 1–2 months of age, marmosets only produce mature sounding contact calls (*Takahashi et al., 2015*; *Zhang and Ghazanfar, 2016*; *Zhang and Ghazanfar, 2018*). One hypothesis is that they simply learn to stop moving when they need to produce a contact call. Doing so would eschew any potential physiological constraints on the production of mature sounding contact calls. In such a scenario, we would predict the number of contact calls produced during movement would decrease, while those produced during periods of immobility would increase. An alternative hypothesis is that as marmosets grow bigger, they become more capable of producing mature sounding contact calls while moving. This outcome would suggest that infants overcome any potential physiological constraints on vocal-locomotor coordination. In such a scenario, we would predict that the number of contact calls produced during movement would increase, while those produced during periods of immobility would decrease.

To test the above hypotheses, we first needed to estimate locomotor activity on a continuous scale. These estimates were extracted from frame-by-frame pixel differences in the video footage of the sessions (*Figure 3—figure supplement 1*). We first summarized the temporal real-time dynamics of locomotor activity surrounding vocal production events (*Figure 3C*). We found that locomotor activity started to *increase* above the 95% threshold of the bootstrap significance test ~5 s before the immature call onsets and then decrease back to inside the 95% threshold ~10 s after call offsets. In contrast, locomotor activity started to *decrease* below the 95% threshold of the bootstrap significance test ~8 s before mature contact call onsets and then increase back to inside the 95% threshold ~10 s after call offsets. A LMM confirmed that the average locomotor activity was higher during immature contact calls as compared to mature contact calls (n = 9609 calls, β ±SE = −0.07±0.01, $t$ = 5.19, p<0.0001; Appendix 1.7). When these temporal dynamics were mapped across postnatal days, we found that locomotor activity during immature contact call production remained around or above the randomized expected levels of locomotor activity. In contrast, early in postnatal life, mature contact call production occurred when locomotor activity was below the 95% threshold of the bootstrap significance test; however, their production gradually became coordinated with increased levels of locomotor activity (*Figure 3D*). Locomotor activity during mature contact calls was higher during late development (PND 52–61) as compared to early development (PND 1–10) (Early = 914 calls; Late = 932 calls; β ±SE = 0.04±0.01, $t$ = 3.05, p=0.0056; Appendix 1.8).

These data therefore support the alternative hypothesis, that the potential constraints of producing mature contact calls during movements are mitigated as the infants get older. Thus, even though postural-locomotor maturity does not appear to be a prerequisite for vocal development (*Figure 2*), vocal-locomotor coordination is still an important component of marmoset monkey motor development. This finding then raises the question of how real-time fluctuations in physiological condition may predict call production and locomotor activity in developing infants. Presumably, infants in elevated states of arousal are more likely to engage in these motor behaviors. The question of whether temporal associations between motor output and arousal levels change over the course of development is addressed next.

## Mature call production and locomotion occurs during elevated arousal levels

We first tested hypotheses about the developmental dynamics of arousal state during marmoset contact call production. One very basic hypothesis is that the production of mature contact calls requires elevated arousal levels more than does the production of immature contact calls (*Teramoto et al., 2017*). Being in an elevated arousal state also means that individuals may have more respiratory power needed to generate mature sounding calls (*Borjon et al., 2016*; *Zhang and*

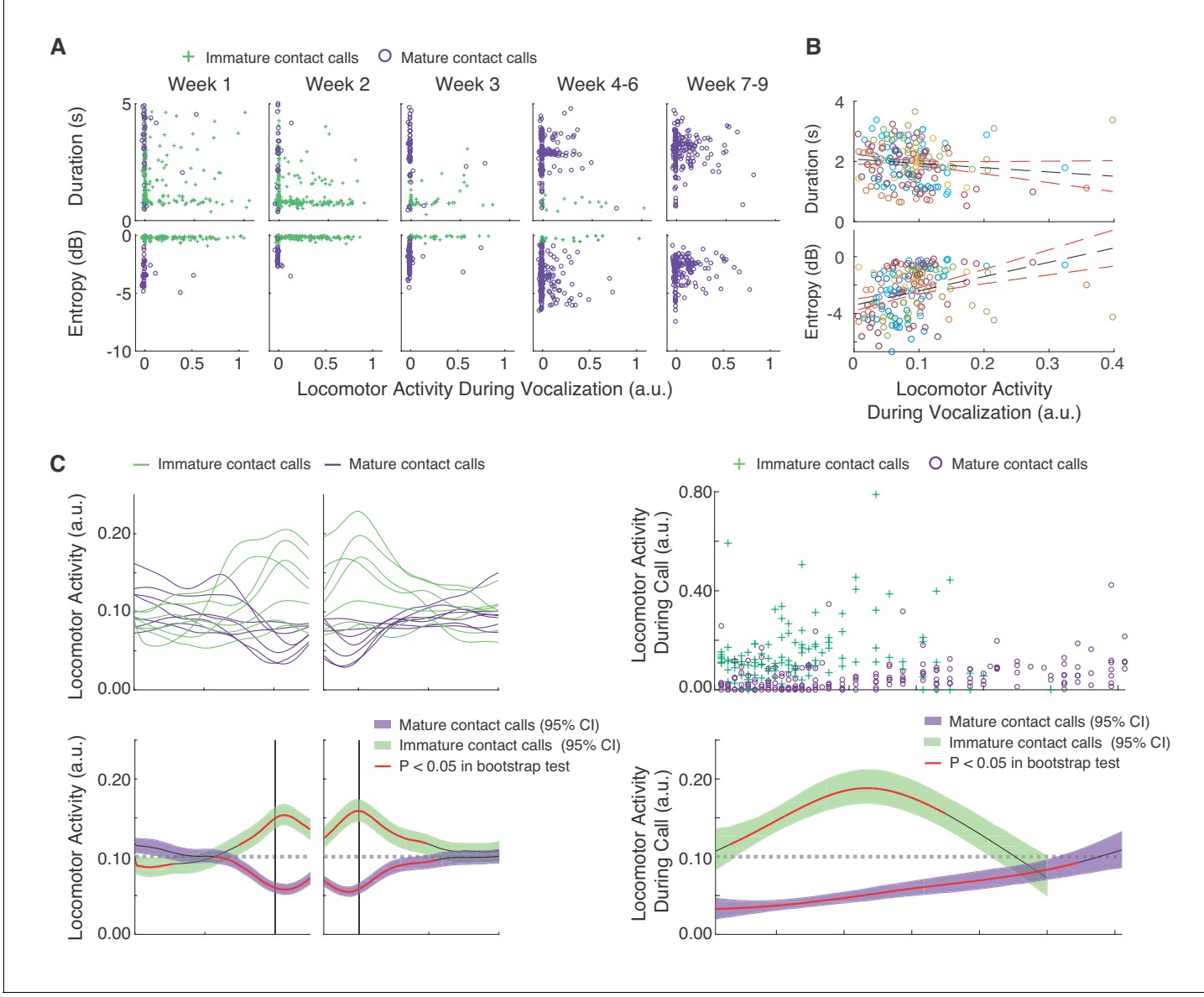

**Figure 3.** Mature contact calls and locomotor activity become increasingly coordinated across development. (A) Plots of contact call acoustic parameters and corresponding locomotor activity in one infant. Green pluses represent immature contact calls (cries), and purple circles represent mature contact calls (phees). (B) Plots of contact call acoustic parameters and locomotor activity levels during contact calls. Points represent subject averages (one color per subject) per day and fit with a linear regression (black dotted line) with 95% confidence intervals (red dotted lines). (C) Locomotor activity from 20 s before to 20 s after contact call production. (C, top) Subject averages (one line per subject) of cubic splines fit to locomotor activity fluctuations surrounding cries (green lines) and phees (purple lines). (C, bottom) Colored cubic splines fit to population data are plotted with a 95% bootstrapped confidence interval (green – cries, purple – phees). Red line indicates observed values outside of the 95% threshold of the bootstrapped significance test. (D) Fluctuations in locomotor activity during contact calls across development. (D, top) Locomotor activity during cries (green pluses) and phees (purple circles) for all observation sessions. (D, bottom) Colored cubic splines fit to population data are plotted with a 95% bootstrapped confidence interval (green – cries, purple – phees). A red line indicates observed values outside of the 95% threshold of the bootstrapped significance test, and a black line indicates observed values inside the 95% threshold.

DOI: https://doi.org/10.7554/eLife.41853.005

The following figure supplement is available for figure 3:

**Figure supplement 1.** Measurement of locomotor activity.
DOI: https://doi.org/10.7554/eLife.41853.006

*Ghazanfar, 2016*). Another hypothesis (that doesn't preclude the first one) is that mature contact calls are more likely to occur during elevated arousal levels earlier in development. This would be consistent with computational models that suggest that younger infants are unable to effectively coordinate the elements of their vocal apparatus to produce mature contact calls and may require enhanced respiratory power to do so (*Takahashi et al., 2015*; *Teramoto et al., 2017*); indeed, there is now empirical support for the link between respiratory power and mature contact call production (*Zhang and Ghazanfar, 2018*). Conversely, a third hypothesis is that mature contact calls are more likely to occur during elevated arousal states later in development than earlier. This could occur if, as the infants get older, their overall motivation changes (i.e., less stressed by isolation) and so higher arousal levels are needed to motivate vocal production. An elevated arousal state may also enable respiratory power and laryngeal tension (*Zhang and Ghazanfar, 2016*); this could lead to better coordination between vocal production and locomotion (*Figure 3C,D*).

We found that mature contact calls occurred during elevated levels of arousal, a pattern that become more pronounced as development proceeded. A summary of the real-time dynamics of heart rate fluctuations in the 10 s before to 10 s after vocal production revealed that marmosets produced immature contact calls when heart rate percentiles dropped below the 95% threshold of the bootstrap significance test (*Figure 4A*). Production of mature contact calls, on the other hand, occurred when heart rate percentiles went above the 95% threshold of the bootstrap significance test. In both cases, changes in arousal levels were 'global', meaning that the change in arousal occurred well before (at least 10 s) the start of the call. A LMM confirmed that heart rate percentiles were higher during mature contact calls than during immature contact calls (n = 6215 calls, $\beta \pm SE = 3.38 \pm 1.45$, $t = 2.32$, p=0.0276; Appendix 1.9). The developmental dynamics of arousal levels during vocal production further supported this main effect (*Figure 4B*). Early in postnatal life, heart rate percentiles during immature contact calls were inside the 95% threshold of the bootstrap significance test, but at around two weeks, heart rates during immature contact calls started to decrease below the 95% threshold. In contrast, mature contact calls were produced when heart rate percentiles were at, or above, the 95% threshold of the bootstrap significance test during the first two months of postnatal life. There was a marginal increase in heart rate percentiles during mature contact calls during late development (PND 52–61) as compared to early development (PND 1–10) (Early = 490 calls; Late = 780 calls; $\beta \pm SE = 6.28 \pm 3.48$, $t = 1.81$, p=0.0795; Appendix 1.10).

We next tested hypotheses about the developmental dynamics of arousal during locomotor activity. One basic hypothesis is that locomotion occurs during elevated arousal states, which is a pattern well supported by studies of heart rate during physically demanding tasks (*Rotstein et al., 2004*; *Baker et al., 2008*). Another hypothesis (that is not mutually exclusive of the first one) is that locomotor activity is more likely to occur during elevated arousal levels earlier in development. One interpretation of this result is that, as with vocal production, younger infants are less able to coordinate their body movements and require enhanced respiratory power to do so. Conversely, a third hypothesis is that locomotor activity is more likely to occur during elevated arousal levels later in development than earlier. As with vocal production, this could occur if, as the infants get older, their overall motivation changes (i.e., less stressed by isolation) and so higher arousal levels are needed to motivate movement. An elevated arousal state may also enable respiratory power and laryngeal tension to enable better coordination between vocal production and locomotion (*Figure 3C,D*).

Like vocal production, we found that locomotor activity occurred during elevated arousal levels, a pattern that appeared to become more pronounced later in development. A summary of the real-time dynamics of heart rate fluctuations in the 10 s before to 10 s after locomotion events revealed that these events occurred when heart rate percentiles were elevated above the 95% threshold of the bootstrap significance test (*Figure 4C*). As with vocal production, this change in arousal was 'global', meaning that the change in arousal happened well before (at least 10 s) the start of movement. The developmental dynamics of arousal levels during locomotor activity further supported this main effect result (*Figure 4D*). Early in postnatal life, heart rate percentiles during locomotor activity were within the 95% threshold of the bootstrap significance test, but continued to increase throughout the first two months of postnatal life. There was a marginal increase in heart rate percentiles during locomotor events during late development (PND 52–61) as compared to early development (PND 1–10) (Early = 752 calls; Late = 566 calls; $\beta \pm SE = 8.12 \pm 4.04$, $t = 2.01$, p=0.0580; Appendix 1.11).

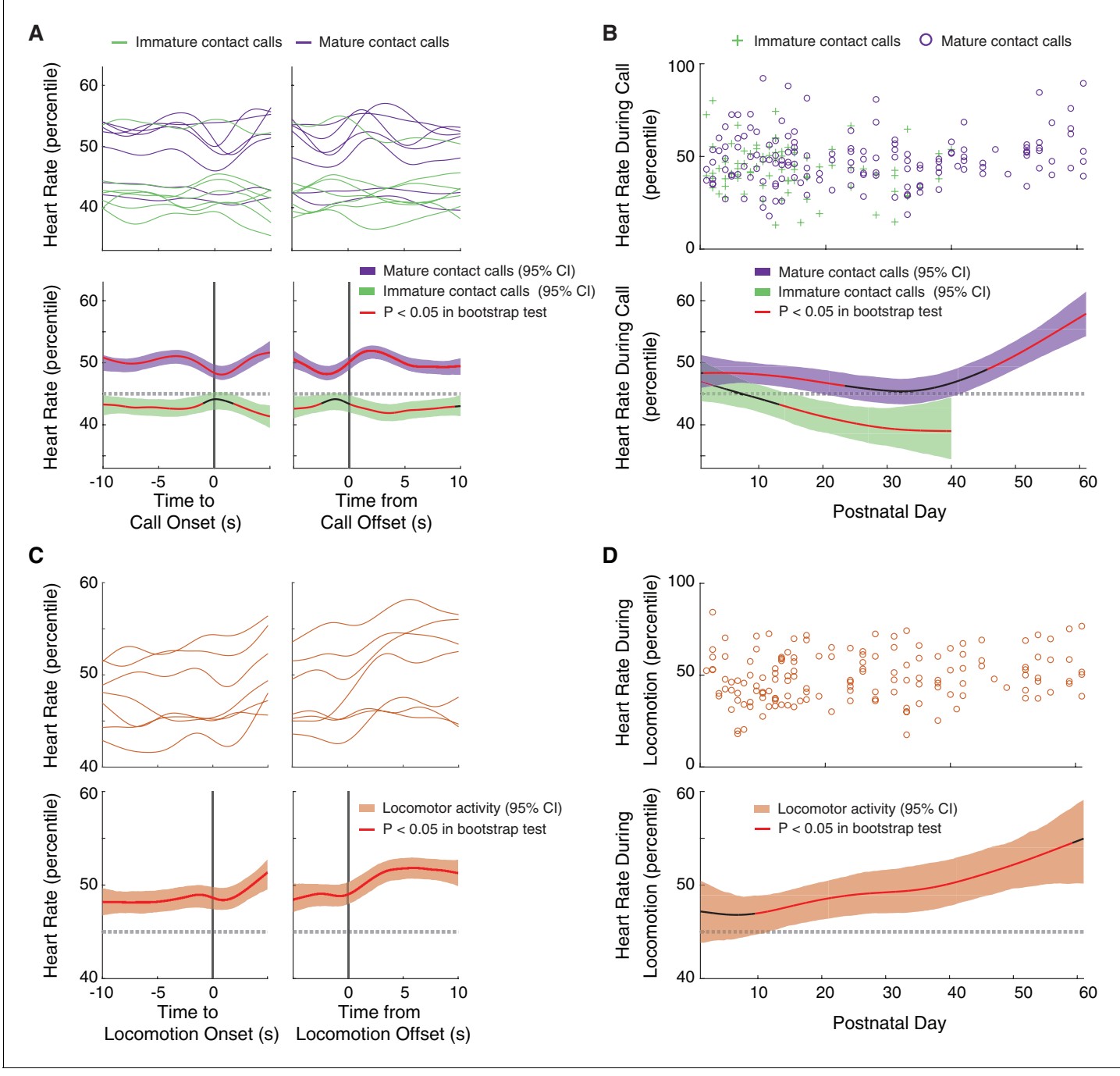

**Figure 4.** Mature contact calls and locomotor activity are more likely to occur during elevated arousal levels later in development. (A) Arousal fluctuations from 10 s before to 10 s after contact call production. (A, top) Subject averages (one line per subject) of cubic splines fit to arousal level fluctuations surrounding cries (green lines) and phees (purple lines). (A, bottom) Colored cubic splines fit to population data are plotted with a 95% bootstrapped confidence interval (green – cries, purple – phees). Red line indicates observed values outside of the 95% threshold of the bootstrapped significance test. (B) Fluctuations in arousal levels during contact calls across development. (B, top) Arousal levels during cries (green pluses) and phees (purple circles) for all observation sessions. (B, bottom) Colored cubic splines fit to population data are plotted with a 95% bootstrapped confidence interval (green – cries, purple – phees). Red line indicates observed values outside of the 95% threshold of the bootstrapped significance test. (C) Arousal fluctuations from 10 s before to 10 s after locomotor activity. (C, top) Subject averages (one line per subject) of cubic splines fit to arousal level fluctuations surrounding locomotor events (orange lines). (C, bottom) A colored cubic spline (orange) fit to population data is plotted with a 95% bootstrapped confidence interval. Red line indicates observed values outside of the 95% threshold of the bootstrapped significance test. (D) Fluctuations in arousal levels during locomotor activity across development. (D, top) Arousal levels during locomotor events (orange circles) for all observation sessions. (D, bottom) A colored cubic spline (orange) fit to population data is plotted with a 95% bootstrapped confidence interval. A red

*Figure 4 continued on next page*

*Figure 4 continued*
line indicates observed values outside of the 95% threshold of the bootstrapped significance test, and a black line indicates observed values inside of the 95% threshold.
DOI: https://doi.org/10.7554/eLife.41853.007
The following figure supplement is available for figure 4:
**Figure supplement 1.** Measurement of arousal state.
DOI: https://doi.org/10.7554/eLife.41853.008

Our data suggest that mature contact call production and locomotor activity are both associated with elevated levels of arousal, an association that becomes more pronounced with age. As such, arousal state may be an important predictor of whether infant marmosets coordinate these different motor outputs. We tested this idea next.

## Coordination of mature contact call production with locomotor activity occurs during elevated levels of arousal

Over the course of infant marmoset development, both locomotor activity and arousal levels predicted whether an infant produces a mature contact call over an immature call. Low levels of locomotor activity support mature contact call production early in development, and elevated arousal levels support mature contact call production later in development. The missing piece of the puzzle is whether there is a connection between vocal-locomotor coordination and arousal state. Here, we test, and find support for, the hypothesis that arousal levels during mature contact call production are elevated when marmosets are moving around (*Figure 5A*). Locomotor activity during mature contact call production was positively associated with heart rate percentiles (n = 3966 calls, $\beta \pm SE = 9.66 \pm 4.05$, $t = 2.39$, p=0.0208; Appendix 1.12; *Figure 5B*). In other words, this result suggests that individuals in an elevated arousal state are better able to coordinate mature vocal production with locomotion. Specifically, this positive association characterized infants that were one month old (1–30 days; n = 1705 calls, $\beta \pm SE = 16.32 \pm 6.60$, $t = 2.47$, Bonferroni-Holm adjusted p=0.0382; Appendix 1.12), but not two months old (31–61 days; n = 2261 calls, $\beta \pm SE = 2.63 \pm 5.24$, $t = 0.50$, Bonferroni-Holm adjusted p=0.6220; Appendix 1.12). This means that a positive association between locomotor activity and heart rate percentiles was not simply a by-product of age. Instead, there may be a particularly robust relationship between locomotor activity and heart rate during early infanthood when individuals are transitioning from producing immature to mature contact calls.

## Discussion

Vocal development is a dynamic process that involves the interaction of multiple systems, from the biomechanics and muscles of the vocal apparatus to the nervous system and the social environment (*Thelen, 1991*; *Teramoto et al., 2017*). Using developing marmoset monkeys, we sought to understand how these processes related to vocal output are influenced by other systems of motor behavior, specifically posture and locomotion. First, we examined when the transition from immature to mature calls occurs relative to the transition from immature to mature body postures and locomotion. Second, we examined the putative temporal coordination of vocal production with locomotor activity, and whether such coordination changes over the course of development. Finally, we investigated whether fluctuating arousal levels (estimated from heart rate) predicts vocal and locomotor output. We found that marmoset monkey vocalizations develop sooner than postural-locomotor skills, that locomotor activity gradually becomes coordinated with the production of mature-sounding contact calls, and that this vocal-locomotor coordination occurs during elevated levels of arousal.

### Head-to-tail sequence of development

The development of the vocal and postural-locomotor systems in marmoset monkeys, at a first approximation, seems to follow similar trajectories (*Wang et al., 2014*; *Takahashi et al., 2015*). By investigating both motor systems in the same individuals longitudinally, however, we showed that vocal behavior matures sooner than either posture or locomotion. Marmosets transitioned to producing a higher proportion of mature-sounding contact calls (phees) than immature-sounding

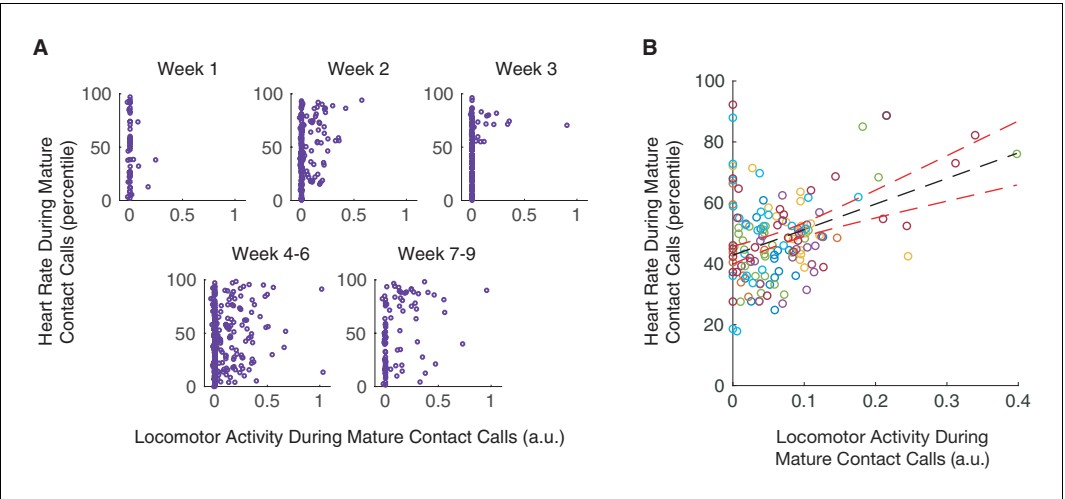

**Figure 5.** Coordination of mature contact calls with locomotor activity occurs during elevated arousal levels. (**A**) Plots of locomotor activity and heart rate percentiles during contact calls in one infant. Purple circles represent mature contact calls (phees). (**B**) Plot of heart rate percentiles and locomotor activity during mature contact calls (phees). Points represent subject averages (one color per subject) per day and fit with a linear regression (black dotted line) with 95% confidence intervals (red dotted lines).
DOI: https://doi.org/10.7554/eLife.41853.009

contact calls (cries) around postnatal day 10, whereas posture and locomotion transitioned to more mature forms around days 18 and 21, respectively. This finding implies that the ability to produce adult-like sounds is not contingent upon advanced postural and locomotor control. These findings also parallel the developmental sequence of human prelinguistic vocal output, posture, and locomotion. Human infants begin producing babbling sounds (consonant-vowel combinations) at around 2 to 4 months (*Vihman, 2014*), unsupported upright sitting around 4 to 5 months (*Adolph, 2008*), and walking without support around 11 months (*Adolph, 2008*). Our data suggest that, like humans and other animals (*Starck and Ricklefs, 1998*; *Muir, 2000*; *Adolph, 2008*), marmoset monkey motor development takes the form of a head-to-tail sequence.

Why do marmoset monkey and human infants both transition to producing more adult-like sounds before they transition to sitting in upright postures and walking? One explanation is that an initial investment in the vocal system allows infants to elicit caregiver attention (e.g., carrying and food sharing) more effectively, thereby prolonging the amount of time they need to develop their locomotor autonomy. This explanation makes sense given the developmental strategy (i.e., altricial) and social system (i.e., cooperative breeding) of humans and marmosets (*Snowdon, 1996*; *Hrdy, 2007*; *Solomon and French, 2007*; *Burkart et al., 2009*; *Lukas and Clutton-Brock, 2012*). For altricial species like marmoset monkeys (relative to other nonhuman primates), early development is an energetically costly period because individuals are unable to fully regulate their body temperature, and yet, altricial infants must invest energy to grow at a high rate and refine their motor skills (*Rosenblatt, 1976*; *Case, 1978*; *Derrickson, 1992*; *Blumberg and Sokoloff, 1998*; *Starck and Ricklefs, 1998*; *Muir, 2000*; *Blumberg, 2001*; *Schilling, 2005*; *Price and Dzialowski, 2018*). In marmosets and humans, locomotor and physiological constraints (such as control of arousal levels) may be overcome by receiving care and physical contact from both maternal and non-maternal adults (*Case, 1978*; *Snowdon, 1996*; *Hrdy, 2007*); infants elicit such contact and care by producing vocalizations (*Locke, 2006*; *Zuberbühler, 2012*; *Ghazanfar and Takahashi, 2017*). And yet, not all vocalizations are created equal. Previous work in marmosets and humans suggests that more adult-like sounds elicit caregiver attention better than do immature sounds (*Gros-Louis et al., 2006*; *Takahashi et al., 2016*). By investing first in developing mature-sounding contact calls, infants may be 'buying the time' they need to learn how to move about independently in their environment.

## Locomotion as a potential constraint on vocal development

The ability to coordinate biomechanical features of the vocal system – breathing, thoracic pressure and vocal fold tension – is critical for producing species-typical vocalizations (*MacLarnon and Hewitt, 1999*; *Maclarnon and Hewitt, 2004*; *Takahashi et al., 2015*; *Zhang and Ghazanfar, 2016*; *Teramoto et al., 2017*). In developing marmosets, a mismatch between biomechanical dynamics of the vocal system is thought to generate immature cries instead of mature contact calls (*Teramoto et al., 2017*). Our study indicates that locomotor activity is one force that can potentially disrupt vocal production. Higher levels of body movements co-occurred with the production of immature-sounding contact calls (i.e., those with a short duration and high entropy), while lower levels of movement co-occurred with mature-sounding contact calls with long durations and lower entropy. These results are consistent with what is known about how movement affects respiration. Vigorous motor activities, like running, speed up breathing cycles (*Wasserman et al., 1973*; *Bramble and Carrier, 1983*), resulting in articulation deficits (*Sundberg et al., 1991*; *Price et al., 2006*; *Baker et al., 2008*; *Orlikoff, 2008*). The finding that calls produced during movement were shorter and noisier indicates that very young marmosets lacked adequate respiratory power (*Zhang and Ghazanfar, 2018*).

## Arousal levels during vocal-locomotor coordination

A unique aspect of our study design is that we could test how real-time fluctuations in arousal related to vocal production, locomotion, and the coordination of these different motor outputs. Infant marmosets produced mature-sounding contact calls during elevated arousal levels and immature ones during low arousal levels, a finding that is consistent with previous work on infant and adult marmosets (*Borjon et al., 2016*; *Zhang and Ghazanfar, 2016*). Similar to mature-sounding contact calls, infant marmosets also tended to engage in locomotion during elevated arousal states. Moreover, real-time variability in arousal state indicated whether mature-sounding contact calls co-occurred with locomotor activity. Infant marmosets exhibited elevated arousal levels when mature-sounding contact calls were produced during movement, and decreased arousal levels when such calls were produced during periods of immobility. These results are consistent with the hypothesis that elevated arousal may help to overcome physiological demands (e.g., respiratory power) of producing vocalizations while moving. Similar associations have been observed in humans engaged in physically demanding tasks (*Rotstein et al., 2004*; *Baker et al., 2008*). For example, heart rate increases at a faster rate during sustained exercise in adults who are engaged in a speech task than in adults engaged in a non-speech task (*Baker et al., 2008*). Our study design precludes testing causality but does suggest that arousal state is a key player in the coordination between vocal and locomotor systems during development.

The inhibition of motor activity during mature contact call production shifted over the course of development. That is, by the time that infant marmoset monkeys stopped producing immature-sounding calls, they no longer showed decreased movement during mature contact call productions. This developmental shift suggests that marmosets gradually improved their ability to coordinate locomotor behaviors with vocal production. From a biomechanical perspective, this improvement suggests that marmosets acquired the ability to better control the vocal apparatus during movement. From a neural perspective, this improvement suggests that more experience with vocal behaviors and/or locomotion leads to better coordination between these motor systems. In either case, arousal state appears to be the common currency by which vocal-motor coordination emerges. One intriguing possibility is that these shifting dynamics of the autonomic nervous system create the scaffolding by which mature social behavior can emerge (*Porges and Furman, 2011*).

Despite being identified as a critical line of inquiry over 30 years ago (*Tipps et al., 1981*; *Yingling, 1981*), to date, only a handful of empirical studies investigated infant vocal-locomotor coordination (*Ejiri and Masataka, 2001*; *Fagan and Iverson, 2007*; *Abney et al., 2014*; *Berger et al., 2017*). Our study using marmoset monkeys as a model for developmental processes represents one of the first to integrate longitudinal and second-by-second timescales to investigate vocal development from a 'whole-body' perspective. We believe that this timescale integration is key to characterizing the fine-grained dynamics that dictate how mature vocalizations emerge. Trade-offs in vocal-locomotor coordination is a potential dynamic that needs to be reckoned with as individuals grow

up. By concurrently measuring heart rate, we show that processes related to autonomic arousal may enable individuals to cope with this trade-off.

## Materials and methods

### Subjects and housing

The subjects used in this study were seven (three females) infant common marmosets (*Callithrix jacchus*) from three different parental pairs (two sets of twins and one set of triplets,<2 months old). Subjects were born in captivity and lived with their family groups (mother, father and siblings). The colony room was maintained at a temperature of approximately 27° C with 50–60% relative humidity, and a 12:12 hr light-dark cycle. All subjects had ad libitum access to water and were supplied daily with standard commercial chow supplemented with fruit and vegetables.

### Experimental design

The experimental protocol follow methods previously described (*Borjon et al., 2016*; *Zhang and Ghazanfar, 2016*). Infant marmosets were separated from their parents and placed in a testing box in an experiment room. The triangular, prism-shaped testing boxes were made of Plexiglas and wire (0.30 m x 0.30 m x 0.35 m). All observation sessions were conducted during daylight hours between postnatal day 1 to 61, and each observation session lasted 10 minutes. Subjects participated in a total of 220 observation sessions across the first two months of life (Subjects 1–7: 29, 29, 34, 34, 31, 31, 32). All experimental procedures were performed in compliance with the guidelines of the Princeton University Institutional Animal Care and Use Committee.

### Vocal behavior data collection

Undirected vocalizations (i.e., socially isolated) were recorded using a Sennheiser MKH416-P48 microphone suspended 0.9 m above the testing box. The microphone signal was sent to a Mackie 402-VLZ3 line mixer whose output was relayed to a Plexon Omniplex and PC computer. We used the same custom MATLAB software established in previous research for computationally defining and segmenting infant marmoset vocalizations (*Takahashi et al., 2015*; *Zhang and Ghazanfar, 2016*). A researcher manually verified which calls were one of two types of contact calls – cries and phees (*Figure 1A*). As described previously (*Takahashi et al., 2015*), cries are contact calls that have a short duration and noisy spectral properties (i.e., high Wiener entropy); phees are contact calls that have a longer duration and tonal spectral properties (i.e., low Wiener entropy). We recorded audio for 192 of the 220 observation sessions (Subjects 1–7: 28, 29, 21, 20, 31, 31, 32) for a total of 10,956 calls (Subjects 1–7: 1,100, 801, 1,604, 1,021, 1,657, 2,199, 2,574). A custom-made MATLAB routine calculated two main acoustic properties for each call, the duration and Wiener entropy (*Takahashi et al., 2016*). Wiener entropy is a non-positive number that is calculated by taking the logarithm of the ratio between the geometric and arithmetic means of the values of the power spectrum for different frequencies (*Tchernichovski et al., 2000*). Wiener entropy represents the broadband properties of a signal's power spectrum in which the closer the signal is to white noise, the higher (closer to zero) the entropy value.

### Postural and locomotor behavior data collection

Postural-locomotor behavior was video recorded at 30 fps with a Plexon Cineplex. We recorded video for 215 of the 220 observation sessions and identified a total of 3195 instances of specific postural-locomotor behaviors (Subjects 1–7: 662, 635, 414, 574, 207, 427, 276). We manually scored the recorded videos to identify the onset and offset of behaviors using BORIS, an open-source event-logging software for video coding (*Friard and Gamba, 2016*). As the video frame rate is at 30 Hz, the onset and offset of behaviors had a maximum resolution of 1/30 s. Our definitions of postural-locomotor behaviors were based on prior literature (*Wang et al., 2014*). During video coding, frames without a clear posture or locomotion described in our ethogram were not assigned a behavior.

Postural behaviors were defined as instances that infants spent repositioning itself: forelimb support, hanging, hindlimb support, raising head, and righting reflex (*Figure 1B*). The righting reflex is when infants re-establish their body orientation so that their hands and feet are on the ground; head

raising is when infants lift their head off the ground and look forward or up; forelimb support is when infants sit on the ground with their hands touching the ground (or cage); hindlimb support is when infants sit on the ground with their hands off the ground (or cage); hanging is when infants grasp the cage so that their hands and feet do not touch the ground. Locomotor behaviors were defined as instances when infants traversed the testing box: crawling, digging, jumping, climbing, and walking (*Figure 1C*). Crawling is when infants move forward on the ground with their stomach touching the ground; digging is when infants move their hands back and forth across the ground; jumping is when infants push themselves off the ground or cage to move from one location to another; climbing is when infants traverse across the cage; walking is when infants traverse across the ground in a standing orientation.

Continuous variability in locomotor activity (i.e., body movement) was assessed by investigating pixel differences in the video data (*Figure 3—figure supplement 1*), following methods used in earlier research (*Borjon et al., 2016*). Each video recording was split into segments of 30 frames (each 640 vs 400 pixels). We took the absolute difference of RGB values between the first and last frame of every second and divided by the total number of pixels. This value corresponded to the average difference in luminescence per pixel per second. A higher value indicates more pixel difference, signifying movement. Because absolute levels of locomotion differ across individuals and ages (e.g., movement of larger individuals causes larger pixel differences), it was necessary to re-scale locomotor activity levels. To do this, we converted all 1 s movement values to binary values with a 90th percentile threshold. Then, we use csaps in MATLAB to fit a cubic spline (smoothing parameter of 0.10) to the binary values in each 10 min observation session. In other words, locomotor activity ranged between zero (immobile) and one (mobile) so that comparisons could be made across individuals and ages.

## Electrocardiography data collection

To quantify arousal fluctuations, we recorded heart rate for 149 of the 220 observation sessions (Subjects 1–7: 16, 14, 21, 19, 26, 25, 28). To record electrocardiographic (ECG) signal, we used two pairs of Ag-AgCl surface electrodes (Grass Technology) (*Figure 4—figure supplement 1*). Tethered electrodes were sewn into a soft elastic band, which was clasped around the animal's thorax. One pair of electrodes was positioned on the dorsal thorax, and the other pair was positioned on the ventral thorax. We applied ECL gel on the surface of each electrode to improved signal-to-noise ratio. Infants were shaved around the thorax if needed. Each pair of electrodes was differentially amplified (x250) with the resulting signal sent to the Plexon Omniplex, where it was digitized at 40 kHz and sent to a personal computer for data acquisition. The strength of heart rate signals varies throughout observation sessions as animal movement alters the positioning of surface electrodes. As such, we chose the channel with the largest signal-to-noise ratio on a session-by-session basis. We manually identified and isolated motion artifacts or signal cutoffs. To minimize bias, we did the following for each observation session: signals from both ECG channels were divided into 10 s segments and signal pairs were presented in random order for visual inspection. Regions exhibiting signal loss were replaced with NaNs.

Following the visual screening, we down-sampled data from 40 kHz to 1,500 Hz to extract the cardiac signals. These signals were high-pass filtered at 15 Hz to preserve the rapid waveform of the heartbeat. The resulting signal was notch filtered at 60 Hz. Heart beats were detected using an adaptive threshold of 1 s duration to find cardiac spikes greater than the 95th percentile of the amplitude at each second of the signal. Occasionally, an artificial spike close to the actual heartbeat would be detected or a heartbeat would be missed. As such, we set inter-spike interval thresholds of 100 ms (600 beats/min) to 400 ms (150 beats/min), which are thresholds used in a previous study of marmoset autonomic activity (*Borjon et al., 2016*). If an inter-spike interval was less than 100 ms, the two spikes were substituted with a single spike located at the midpoint. If an inter-spike interval exceeded 400 ms, we replaced the signal with a NaN. To calculate heart rate, we constructed a binary series of heartbeat counts and convolved the resulting series with a 1 s Gaussian window. We only used heart rate data from observation sessions during which heart rate could be detected at least 50% of the time. Because heart rate can differ across individuals and ages, it was necessary to re-scale heart rate fluctuations so that comparisons could be made across individuals and ages. To do so, we converted all 1 s heart rate values to percentiles for each 10 min observation session (i.e., heart beat fluctuations were centered around the 50th percentile). In other words, heart rate

percentile values ranged from zero (lowest heart rate level) to 100 (highest heart rate level) within each 10 min observation session.

## Data analysis

All analyses were carried out in MATLAB (version R2019a) and R (version 3.5.3). To determine which motor behaviors were categorized as 'immature' and 'mature', we calculated the proportion of vocal, postural, or locomotor time engaged in specific behaviors. We used a series of linear mixed effect models (LMM, 'lmer' of the R package 'lme4'; *Bates et al., 2015*) to test whether the proportion of time engaged in these behaviors changed based on postnatal day. We used the 'lmerTest' R package to determine the significance of the coefficients (*Kuznetsova et al., 2017*). In the LMMs to examine how vocal behavior changes across development, the dependent variable was the proportion of total vocal time (per daily observation session) engaged in a specific vocalization (cry or phee), the fixed effect was postnatal day, and the random effect was the infant subject (LMM equation: proportion of time ~day + (day|subject)). In the LMMs to examine how postural behavior changes across development, the dependent variable was the proportion of total postural time (per daily observation session) engaged in a specific posture (forelimb support, hanging, hindlimb support, raising head, or righting reflex), the fixed effect was postnatal day, and the random effect was infant subject (LMM equation: proportion of time ~day + (day|subject)). In the LMMs to examine how locomotor behavior changes across development, the dependent variable was the proportion of total locomotor time (per daily observation session) engaged in a specific posture (climbing, crawling, digging, jumping, or walking), the fixed effect was postnatal day, and the random effect was infant subject (LMM equation: proportion of time ~day + (day|subject)). We applied the Bonferroni-Holm method to correct for issues of multiplicity within each behavior type (vocal, postural, or locomotor behaviors), resulting in adjusted p-values with an alpha threshold level of 0.05. We report detailed outcomes of these regression models (e.g., model formulas, random effect variance, regression coefficients, standard errors, t-values, and p-values) in Appendix 1.1-1.3. Immature behaviors were those whose use decreased across development, and mature behaviors were those whose use increased across development. We used these immature-mature categories to calculate 'maturity indices' per session. This index ranged from 0 to 1 and was calculated as follows,

$$Maturity\ Index = \frac{m}{m + im}$$

where $m$ represents the percent of time spent engaged in mature behavior and $im$ represents the percent of time engaged in immature behavior. A maturity index value less than 0.5 means that an individual produced more immature behavior, and a value greater than 0.5 means that an individual produced more mature behavior. Cubic splines (MATLAB csaps function) were fit to individual data (smoothing parameter of 0.03) and population data (smoothing parameter of 0.01) to determine the developmental trajectories of vocal, postural and locomotor maturity indices. Then, we used a logistic generalized linear mixed effect model (GLMM, 'glmer' of the R package 'lme4'; *Bates et al., 2015*) to test whether maturation time courses differed between vocal and postural-locomotor behaviors. We used the 'lmerTest' R package to determine the significance of the coefficients (*Kuznetsova et al., 2017*). In this GLMM, the dependent variable was maturity index (per daily observation session, per behavior type—vocal, postural, locomotor), the fixed effect was postnatal day, and the random effect was infant subject (GLMM equation: maturity index ~behavior type +day + (behavior type|subject) + (day|subject)). In Appendix 1.4, we report detailed outcomes of this regression model (e.g., model formulas, random effect variance, regression coefficients, standard errors, t-values, and p-values), as well as the outcomes of this model per infant subject.

We sought to understand the real-time and developmental dynamics between vocal production and locomotor activity, as well as between arousal fluctuations and vocal-locomotor behavior. Then, to visualize the real-time dynamics of locomotion during call production, we extracted locomotor activity from −20 to 5 s surrounding call onsets and −5 to 20 s surrounding call offsets. To visualize the real-time dynamics of arousal fluctuations surrounding vocal-locomotor events, we extracted heart rate percentiles from −10 to 5 s surrounding call (or locomotor activity) onsets and −5 to 10 s surrounding call (or locomotor activity) offsets. To summarize the real-time dynamics for individual sessions, we fit a cubic spline to the session data (MATLAB csaps, smoothing parameter of 0.1), and then we fit a population spline to all session splines (smoothing parameter of 0.3). To visualize

developmental dynamics, we used the average locomotor activity (or heart rate percentile) during each call (or locomotor activity event) to calculate the mean locomotor activity (or heart rate percentile) per individual session. Then, we fit cubic splines (smoothing parameter of 0.0001) to individual sessions to model the change in the population data across postnatal days.

To test the statistical significance of real-time and developmental dynamics, confidence intervals for the population splines were generated by randomly resampling (10 samples per session) from the signals used to generate individual session splines (real-time dynamics) or session averages (developmental dynamics). We fit a cubic spline to the resampled data to generate a population spline, and repeated the process 1000 times. The 95% confidence interval corresponds to the 2.5th and the 97.5th percentiles of the resampled population splines. To determine whether the real-time or developmental dynamics of locomotor activity and arousal fluctuations were significantly different from null expectations, we performed bootstrapped significance tests. For each session, we scrambled the order of call durations and inter-call interval durations. This allowed us to choose random segments equivalent in the number and length to the calls produced in that session while maintaining naturalistic spacing between the calls. We fit a cubic spline to session splines (for temporal analysis) or session averages (for developmental analysis) to generate bootstrapped population splines. We repeated this process 1000 times. The 95% threshold for significance corresponds to the 2.5th and 97.5th percentiles of the bootstrapped population splines. With this bootstrap procedure, we are taking into account data variability due to day, subject and call timing on the statistical estimates.

To complement our bootstrap procedure, we used a series of linear mixed effect models (LMM, 'lmer' of the R package 'lme4'; *Bates et al., 2015*) to investigate associations between call production, locomotor activity, and arousal fluctuations. We used the 'lmerTest' R package to determine the significance of the coefficients (*Kuznetsova et al., 2017*). We used LMMs to investigate how average locomotor activity during a call predicted call duration and Wiener entropy. In these LMMs, the dependent variable was contact call acoustic parameter (call duration or Wiener entropy), the fixed effects was average locomotor activity (per call), and the random effect was infant subject nested within postnatal day (LMM equation: acoustic parameter ~locomotor activity + (locomotor activity|day/subject)). We applied the Bonferroni-Holmes method to correct for multiplicity issues associated with testing two acoustic parameters, resulting in adjusted p-values with an alpha threshold level of 0.05. In Appendix 1.5-1.6, we report detailed outcomes of these regression models (e. g., model formulas, random effect variance, regression coefficients, standard errors, t-values, and p-values), as well as the outcomes of these models per postnatal day and infant subject. To complement the real-time dynamic analyses, we used LMMs to compare locomotor activity and ANS activity between contact call types. For the LMM examining locomotor activity, the dependent variable was the average locomotor activity (per contact call), the fixed effect was contact call type (immature vs. mature), and the random effect was infant subject nested within postnatal day (LMM equation: locomotor activity ~call type + (call type|day/subject)). For the LMM examining ANS activity, the dependent variable was the average heart rate percentile (per contact call), the fixed effect was contact call type (immature vs. mature), and the random effect was infant subject nested within postnatal day (LMM equation: heart rate ~call type + (call type|day/subject)). In Appendices 1.7 and 1.9. we report detailed outcomes of these regression models for example model formulas, random effect variance, regression coefficients, standard errors, t-values, and p-values), as well as outcomes of the models per postnatal day and infant subject.

We also used LMMs to complement the developmental dynamics analyses. We ran an LMM to examine developmental changes in locomotor activity during mature contact calls, in which the dependent variable was the average locomotor activity (per contact call), the fixed effect was age group, and the random effect was infant subject nested within postnatal day (LMM equation: locomotor activity ~age group + (age group|day/subject)). Age group was split into early development (PND 1–10; infant contact calls transition to sounding more adult-like around PND 10) and late development (PND 52–61). In Appendix 1.8, we report detailed outcomes of this regression model, as well as outcomes for each infant subject separately. We ran an LMM to examine developmental changes in ANS activity during mature contact calls, in which the dependent variable was the average heart rate percentile (per contact call), the fixed effect was age group, and the random effect was infant subject nested within postnatal day (LMM equation: heart rate ~age group + (age group|day/subject)). In Appendix 1.10, we report detailed outcomes of this regression model, as well as

outcomes for each infant subject separately. We also ran an LMM to examine developmental changes in ANS activity during locomotor activity events, in which the dependent variable was the average heart rate percentile (per locomotor event) the fixed effect was age group, and the random effect was infant subject nested within postnatal day (LMM equation: heart rate ~age group + (age group|day/subject)). In Appendix 1.11, we report detailed outcomes of this regression model, as well as outcomes for each infant subject separately.

Finally, we used a LMM to investigate how ANS activity during mature contact calls is associated with vocal-locomotor coordination. The dependent variable of this LMM was average heart rate percentile (per mature contact call), the fixed effect was average locomotor activity (per mature contact call), and the random effect was infant subject nested within postnatal day (LMM equation: heart rate ~locomotor activity + (locomotor activity|day/subject)). We also ran this model on two separate subsets of the data, the first month of development (postnatal day 1–30) and the second month of development (postnatal day 31–61). We applied a Bonferroni-Holmes method to correct for issues of multiplicity due to testing different time frames separately, resulting in adjusted p-values with an alpha threshold level of 0.05. In Appendix 1.12, we report detailed outcomes of these regression models (e.g., model formula, random effect variance, regression coefficients, standard errors, t-values, and p-values), as well as the outcomes for each postnatal day and subject separately.

## Acknowledgements

We thank Lauren Kelly for her help with experiments and careful reading and editing of an earlier version of this manuscript. We thank Steve Phelps and Jon Sakata for their feedback on an earlier version of the manuscript. We also thank Yisi Zhang for advice on analyzing heart rates. This work was supported by NIH T32MH065214 (MLG), NSF Graduate Fellowship (JIB), and NIH R01NS054898 (AAG).

## Additional information

### Funding

| Funder | Grant reference number | Author |
|---|---|---|
| National Institute of Mental Health | T32 MH065214 | Morgan L Gustison |
| National Institute of Neurological Disorders and Stroke | R01 NS054898 | Asif A Ghazanfar |
| National Science Foundation | | Jeremy I Borjon |

The funders had no role in study design, data collection and interpretation, or the decision to submit the work for publication.

### Author contributions

Morgan L Gustison, Data curation, Formal analysis, Investigation, Writing—original draft; Jeremy I Borjon, Data curation, Investigation, Writing—review and editing; Daniel Y Takahashi, Conceptualization, Data curation, Validation, Investigation, Methodology, Writing—review and editing; Asif A Ghazanfar, Conceptualization, Supervision, Funding acquisition, Validation, Writing—original draft, Writing—review and editing

### Author ORCIDs

Morgan L Gustison (ID) https://orcid.org/0000-0002-1162-8966
Jeremy I Borjon (ID) http://orcid.org/0000-0001-9114-3362
Daniel Y Takahashi (ID) https://orcid.org/0000-0003-4972-001X
Asif A Ghazanfar (ID) https://orcid.org/0000-0003-1960-7470

### Ethics

Animal experimentation: This study was performed in strict accordance with the recommendations in the Guide for the Care and Use of Laboratory Animals of the National Institutes of Health. All of

the animals were handled according to approved institutional animal care and use committee (IACUC) protocols (#1908-18) of Princeton University.

## Decision letter and Author response

Decision letter https://doi.org/10.7554/eLife.41853.015
Author response https://doi.org/10.7554/eLife.41853.016

# Additional files

## Supplementary files

• Transparent reporting form
DOI: https://doi.org/10.7554/eLife.41853.010

## Data availability

As part of this full submission, we have uploaded our data to Dryad (https://doi.org/10.5061/dryad.cp75158).

The following dataset was generated:

| Author(s) | Year | Dataset title | Dataset URL | Database and Identifier |
|---|---|---|---|---|
| Gustison ML, Borjon JI, Takahashi DY, Ghazanfar AA | 2019 | Data from:Vocal and locomotor coordination develops in association with arousal state | https://doi.org/10.5061/dryad.cp75158 | Dryad Digital Repository, 10.5061/dryad.cp75158 |

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

## Appendix 1

DOI: https://doi.org/10.7554/eLife.41853.011

### Linear Regression Models

We report below the model formulas, random effect variance, estimated regression coefficients, standard errors, test statistics and p values of the models reported in the main text.

## 1.1 Linear mixed models to predict the proportion of vocal time spent producing specific contact calls (cries or phees) from postnatal day. Results are associated with *Figure 2A*

### 1.1.1 Results for cries

Formula: Cry ~Day + (Day | Subject)
    Data: BehaviorProportions
    Random effects:

| Groups | Name | Variance | Std.dev. | Corr |
|---|---|---|---|---|
| Subject | (Intercept) | 8.116e-02 | 0.28489 | |
| | Day | 8.116e-02 | 0.00446 | −1.00 |
| Residual | | 3.973e-02 | 0.19931 | |

Number of obs: 192, groups: Subject, 7
Fixed effects:

| | Estimate | Std. error df | | T value | Pr(>|t|) |
|---|---|---|---|---|---|
| (Intercept) | 0.592573 | 0.110338 | 4.640188 | 5.371 | 0.00377 |
| Day | −0.012849 | 0.001867 | 4.879883 | −6.883 | 0.00109 |

### 1.1.2 Results for phees

Formula: Phee ~ Day + (Day | Subject)
    Data: BehaviorProportions
    Random effects:

| Groups | Name | Variance | Std.dev. | Corr |
|---|---|---|---|---|
| Subject | (Intercept) | 6.786e-02 | 0.26049 | |
| | Day | 1.641e-05 | 0.00405 | −1.00 |
| Residual | | 3.992e-02 | 0.19981 | |

Number of obs: 192, groups: Subject, 7
Fixed effects:

| | Estimate | Std. error | Df | T value | Pr(>|t|) |
|---|---|---|---|---|---|
| (Intercept) | 0.407145 | 0.101371 | 6.040726 | 4.016 | 0.006891 |
| Day | 0.012857 | 0.001729 | 6.337850 | 7.434 | 0.000235 |

## 1.2 Linear mixed models to predict proportion of posture time spent engaged in specific postures (forelimb support, hanging, hindlimb support, raising head, or righting reflex) from postnatal day. Results are associated with *Figure 2A*

### 1.2.1 Results for forelimb support

Formula: ForelimbSupport ~ Day + (Day | Subject)
    Data: BehaviorProportions
    Random effects:

| Groups | Name | Variance | Std.dev. | Corr |
|---|---|---|---|---|
| Subject | (Intercept) | 2.555e-02 | 0.159854 | |
| | Day | 1.414e-05 | 0.003761 | 0.5 |
| Residual | | 1.250e-01 | 0.353497 | |

Number of obs: 201, groups: Subject, 7
Fixed effects:

| | Estimate | Std. error | Df | T value | Pr(>|t|) |
|---|---|---|---|---|---|
| (Intercept) | 0.377378 | 0.072636 | 5.29752 | 5.195 | 0.00294 |
| Day | −0.001024 | 0.002017 | 5.53592 | −0.508 | 0.63123 |

### 1.2.2 Results for hanging

Formula: Hanging ~Day + (Day | Subject)
    Data: BehaviorProportions
    Random effects:

| Groups | Name | Variance | Std.dev. | Corr |
|---|---|---|---|---|
| Subject | (Intercept) | 1.008e-03 | 0.031752 | |
| | Day | 1.030e-07 | 0.000321 | −1.00 |
| Residual | | 7.370e-02 | 0.271470 | |

Number of obs: 201, groups: Subject, 7
Fixed effects:

| | Estimate | Std. error | Df | T value | Pr(>|t|) |
|---|---|---|---|---|---|
| (Intercept) | 0.157781 | 0.033172 | 10.820946 | 4.756 | 0.000621 |
| Day | −0.001833 | 0.001098 | 94.343478 | −1.669 | 0.098491 |

### 1.2.3 Results for hindlimb support

Formula: HindlimbSupport ~ Day + (Day | Subject)
    Data: BehaviorProportions
    Random effects:

| Groups | Name | Variance | Std.dev. | Corr |
|---|---|---|---|---|
| Subject | (Intercept) | 9.179e-05 | 0.009581 | |
| | Day | 7.557e-05 | 0.008693 | −1.00 |

*continued*

| Groups | Name | Variance | Std.dev. | Corr |
|---|---|---|---|---|
| Residual | | 5.857e-02 | 0.242022 | |

Number of obs: 201, groups: Subject, 7
Fixed effects:

| | Estimate | Std. error | Df | T value | Pr(>|t|) |
|---|---|---|---|---|---|
| (Intercept) | −0.036693 | 0.027849 | 92.117003 | −1.318 | 0.19091 |
| Day | 0.01297 | 0.003429 | 5.942058 | 3.782 | 0.00933 |

## 1.2.4 Results for raising head

Formula: RaiseHead ~ Day + (Day | Subject)
    Data: BehaviorProportions
    Random effects:

| Groups | Name | Variance | Std.dev. | Corr |
|---|---|---|---|---|
| Subject | (Intercept) | 3.447e-03 | 0.058707 | |
| | Day | 1.221e-06 | 0.001105 | −1.00 |
| Residual | | 4.122e-02 | 0.203028 | |

Number of obs: 201, groups: Subject, 7
Fixed effects:

| | Estimate | Std. error | Df | T value | Pr(>|t|) |
|---|---|---|---|---|---|
| (Intercept) | 0.116094 | 0.032054 | 7.042929 | 3.622 | 0.0084 |
| Day | −0.002126 | 0.000917 | 12.62578 | −2.319 | 0.0379 |

## 1.2.5 Results for righting reflex

Formula: RightingReflex ~ Day + (Day | Subject)
    Data: BehaviorProportions
    Random effects:

| Groups | Name | Variance | Std.dev. | Corr |
|---|---|---|---|---|
| Subject | (Intercept) | 6.187e-03 | 0.07866 | |
| | Day | 4.616e-06 | 0.002149 | −1.00 |
| Residual | | 9.023e-02 | 0.300387 | |

Number of obs: 201, groups: Subject, 7
Fixed effects:

| | Estimate | Std. error | Df | T value | Pr(>|t|) |
|---|---|---|---|---|---|
| (Intercept) | 0.386506 | 0.045337 | 6.160551 | 8.525 | 0.000124 |
| Day | −0.008066 | 0.001456 | 8.71568 | −5.54 | 0.000405 |

## 1.3 Linear mixed models to predict proportion of locomotor time spent engaged in specific locomotor behaviors (climbing, crawling, digging, jumping, or walking) from postnatal day. Results are associated with *Figure 2A*

### 1.3.1 Results for climbing

Formula: Climb ~Day + (Day | Subject)
    Data: BehaviorProportions
    Random effects:

| Groups | Name | Variance | Std.dev. | Corr |
|--------|------|----------|----------|------|
| Subject | (Intercept) | 4.207e-03 | 0.064858 | |
| | Day | 3.389e-08 | 0.0001841 | 1.00 |
| Residual | | 7.232e-02 | 0.2689296 | |

Number of obs: 191, groups: Subject, 7
Fixed effects:

| | Estimate | Std. error | Df | T value | Pr(>|t|) |
|--|----------|-----------|----|---------|----------|
| (Intercept) | 0.197306 | 0.03965 | 10.026126 | 4.976 | 0.000552 |
| Day | −0.002325 | 0.001109 | 139.768818 | −2.096 | 0.037849 |

### 1.3.2 Results for crawling

Formula: Crawl ~Day + (Day | Subject)
    Data: BehaviorProportions
    Random effects:

| Groups | Name | Variance | Std.dev. | Corr |
|--------|------|----------|----------|------|
| Subject | (Intercept) | 3.671e-02 | 0.191607 | |
| | Day | 1.313e-05 | 0.003624 | −0.83 |
| Residual | | 1.064e-01 | 0.326136 | |

Number of obs: 191, groups: Subject, 7
Fixed effects:

| | Estimate | Std. error | Df | T value | Pr(>|t|) |
|--|----------|-----------|----|---------|----------|
| (Intercept) | 0.84963 | 0.081707 | 4.292132 | 10.398 | 0.000333 |
| Day | −0.016048 | 0.001921 | 4.816723 | −8.355 | 0.000481 |

### 1.3.3 Results for digging

Formula: Dig ~Day + (Day | Subject)
    Data: BehaviorProportions
    Random effects:

| Groups | Name | Variance | Std.dev. | Corr |
|--------|------|----------|----------|------|
| Subject | (Intercept) | 8.017e-04 | 0.028315 | |
| | Day | 6.872e-06 | 0.002621 | −1.00 |

*continued*

| Groups | Name | Variance | Std.dev. | Corr |
|--------|------|----------|----------|------|
| Residual | | 1.135e-02 | 0.106525 | |

Number of obs: 191, groups: Subject, 7
Fixed effects:

| | Estimate | Std. error | Df | T value | Pr(>|t|) |
|--|----------|-----------|-----|---------|---------|
| (Intercept) | −0.032413 | 0.016347 | 8.552887 | −1.983 | 0.0804 |
| Day | 0.003286 | 0.001085 | 6.050216 | 3.028 | 0.0229 |

### 1.3.4 Results for jumping

Formula: Jump ~Day + (Day | Subject)
 Data: BehaviorProportions
 Random effects:

| Groups | Name | Variance | Std.dev. | Corr |
|--------|------|----------|----------|------|
| Subject | (Intercept) | 1.708e-08 | 0.0001307 | |
| | Day | 2.293e-07 | 0.0004789 | −0.30 |
| Residual | | 9.220e-04 | 0.0303643 | |

Number of obs: 191, groups: Subject, 7
Fixed effects:

| | Estimate | Std. error | Df | T value | Pr(>|t|) |
|--|----------|-----------|-----|---------|---------|
| (Intercept) | −0.0025960 | 0.003521 | 18.830 | −0.737 | 0.4619 |
| Day | 0.0005226 | 0.0002206 | 9.152 | 2.369 | 0.0415 |

### 1.3.5 Results for walking

Formula: Walk ~Day + (Day | Subject)
 Data: BehaviorProportions
 Random effects:

| Groups | Name | Variance | Std.dev. | Corr |
|--------|------|----------|----------|------|
| Subject | (Intercept) | 3.903e-03 | 0.0624716 | |
| | Day | 7.360e-07 | 0.0008579 | 0.98 |
| Residual | | 8.263e-02 | 0.2874487 | |

Number of obs: 191, groups: Subject, 7
Fixed effects:

| | Estimate | Std. error | Df | T value | Pr(>|t|) |
|--|----------|-----------|-----|---------|---------|
| (Intercept) | −0.013853 | 0.040833 | 11.37347 | −0.339 | 0.741 |
| Day | 0.014683 | 0.001229 | 18.151564 | 11.951 | 4.87e-10 |

## 1.4 Linear regression models to compare maturity indices between vocal and postural-locomotor behavior

### 1.4.1 Population-level logistic generalized linear mixed model results

Formula: Index ~ as.factor(BehaviorTypeDummy)+Day + (BehaviorTypeDummy | Subject) + (Day|Subject)

Data: MaturityIndices

Random effects:

| Groups | Name | Variance | Std.dev. | Corr |
|---|---|---|---|---|
| Subject | (Intercept) | 0.760731 | 0.87220 | |
| | BehaviorTypeDummy | 1.313062 | 1.14589 | −1.00 |
| Subject.1 | (Intercept) | 0.022107 | 0.14868 | |
| | Day | 0.003983 | 0.06311 | 0.99 |

Number of obs: 519, groups: Subject, 7

Fixed effects:

| | Estimate | Std. error | Z value | Pr(>|z|) |
|---|---|---|---|---|
| (Intercept) | −1.92129 | 0.45544 | −4.219 | 2.46e-05 |
| as.factor(BehaviorTypeDummy)1 | −2.03326 | 0.58452 | −3.479 | 0.000504 |
| as.factor(BehaviorTypeDummy)2 | −2.86305 | 0.97803 | −2.927 | 0.003419 |
| Day | 0.20878 | 0.03299 | 6.329 | 2.47e-10 |

### 1.4.2 Logistic generalized linear model results per Subject

Formula: Index ~ as.factor(BehaviorTypeDummy)+Day

Data: MaturityIndices[MaturityIndices$Subject == i,])

| Subject | N | Fixed effects | Estimate | Std. error | Z value | Pr(>|z|) |
|---|---|---|---|---|---|---|
| 1 | 80 | Vocal vs. Postural | −1.20 | 0.87 | −1.37 | 0.1702 |
| | | Vocal vs. Locomotor | −0.86 | 0.78 | −1.10 | 0.2724 |
| | | Day | 0.22 | 0.06 | 3.86 | 0.0001 |
| 2 | 78 | Vocal vs. Postural | −4.93 | 1.54 | −3.20 | 0.0014 |
| | | Vocal vs. Locomotor | −5.35 | 1.64 | −3.26 | 0.0011 |
| | | Day | 0.36 | 0.10 | 3.56 | 0.0004 |
| 3 | 65 | Vocal vs. Postural | −2.15 | 1.03 | −2.09 | 0.0362 |
| | | Vocal vs. Locomotor | −4.57 | 1.42 | −3.23 | 0.0013 |
| | | Day | 0.15 | 0.04 | 3.55 | 0.0004 |
| 4 | 78 | Vocal vs. Postural | −3.32 | 1.16 | −2.87 | 0.0041 |
| | | Vocal vs. Locomotor | −4.58 | 1.38 | −3.32 | 0.0009 |
| | | Day | 0.25 | 0.07 | 3.64 | 0.0003 |
| 5 | 67 | Vocal vs. Postural | −3.94 | 2.01 | −1.96 | 0.0497 |
| | | Vocal vs. Locomotor | −4.51 | 1.52 | −2.97 | 0.0030 |
| | | Day | 0.21 | 0.07 | 3.19 | 0.0014 |
| 6 | 80 | Vocal vs. Postural | −1.01 | 0.72 | −1.40 | 0.1607 |
| | | Vocal vs. Locomotor | −1.79 | 0.76 | −2.34 | 0.0190 |
| | | Day | 0.11 | 0.02 | 4.42 | <0.0001 |

*continued*

| Subject | N | Fixed effects | Estimate | Std. error | Z value | Pr(>|z|) |
|---------|-----|----------------------|----------|-----------|---------|----------|
| 7 | 71 | Vocal vs. Postural | −0.33 | 0.82 | −0.40 | 0.6917 |
| | | Vocal vs. Locomotor | 0.30 | 0.77 | 0.39 | 0.6989 |
| | | Day | 0.10 | 0.02 | 4.39 | <0.0001 |

## 1.5 Linear regression models to predict contact call duration from locomotor activity. Results are associated with *Figure 3B*

### 1.5.1 Population-level linear mixed model results

Formula: Duration ~ LocomotorActivityMean + (LocomotorActivityMean | Day/Subject)
   Data: CallParameters
   Random effects:

| Groups | Name | Variance | Std.dev. | Corr |
|-------------|----------------------|----------|----------|-------|
| Subject:Day | (Intercept) | 0.4223 | 0.6498 | |
| | LocomotorActivityMean | 1.0254 | 1.0126 | −0.02 |
| Day | (Intercept) | 0.0000 | 0.0000 | |
| | LocomotorActivityMean | 0.1442 | 0.3797 | NaN |

Residual 1.7564 1.3253

Number of obs: 9609, groups: Subject:Day, 188; Day, 38
   Fixed effects:

| | Estimate | Std. error | Df | T value | Pr(>|t|) |
|-----------------------|----------|-----------|-----------|---------|----------|
| (Intercept) | 2.29114 | 0.05021 | 187.37825 | 45.630 | <2e-16 |
| LocomotorActivityMean | −0.55932 | 0.13149 | 36.32386 | −4.254 | 0.000141 |

### 1.5.2 Linear mixed model results per Day

Formula: Duration ~ LocomotorActivityMean + (1|Subject)
   Data: CallParameters[CallParameters$Day == i,]

| Day | N | Estimate | Std. error | T value | Pr(>|t|) |
|-----|-----|----------|-----------|---------|----------|
| 1 | 471 | −0.46 | 0.52 | −0.88 | 0.3794 |
| 2 | 369 | −0.26 | 0.43 | −0.59 | 0.5529 |
| 3 | 345 | 0.01 | 0.42 | 0.02 | 0.9823 |
| 4 | 324 | −0.43 | 0.44 | −0.99 | 0.3245 |
| 5 | 353 | −1.34 | 0.58 | −2.33 | 0.0203 |
| 6 | 403 | −0.34 | 0.39 | −0.87 | 0.3831 |
| 7 | 336 | −0.90 | 0.52 | −1.73 | 0.0846 |
| 8 | 352 | 0.43 | 0.51 | 0.84 | 0.4009 |
| 9 | 379 | −0.51 | 0.32 | −1.57 | 0.1164 |
| 10 | 258 | −1.71 | 0.31 | −5.59 | <0.0001 |
| 11 | 297 | −1.29 | 0.31 | −4.21 | <0.0001 |
| 12 | 336 | −0.64 | 0.33 | −1.91 | 0.0564 |
| 13 | 348 | −1.60 | 0.29 | −5.51 | <0.0001 |

*continued*

| Day | N | Estimate | Std. error | T value | Pr(>|t|) |
|-----|-----|----------|-----------|---------|---------|
| 14 | 250 | −0.21 | 0.29 | −0.74 | 0.4628 |
| 15 | 365 | −0.50 | 0.32 | −1.54 | 0.1243 |
| 16 | 113 | −1.09 | 0.52 | −2.09 | 0.0386 |
| 17 | 241 | −0.59 | 0.36 | −1.65 | 0.1005 |
| 19 | 200 | −2.13 | 0.34 | −6.25 | <0.0001 |
| 21 | 195 | −0.76 | 0.27 | −2.82 | 0.0053 |
| 24 | 217 | −1.93 | 0.28 | −6.87 | <0.0001 |
| 26 | 255 | −1.11 | 0.29 | −3.81 | 0.0002 |
| 28 | 272 | −0.42 | 0.28 | −1.51 | 0.1312 |
| 31 | 323 | −0.52 | 0.30 | −1.73 | 0.0851 |
| 33 | 312 | 0.10 | 0.31 | 0.31 | 0.7555 |
| 35 | 222 | −0.08 | 0.33 | −0.24 | 0.8089 |
| 38 | 251 | −0.65 | 0.35 | −1.83 | 0.0689 |
| 40 | 122 | −1.21 | 0.56 | −2.17 | 0.0318 |
| 42 | 269 | −0.08 | 0.25 | −0.34 | 0.7371 |
| 45 | 259 | −0.43 | 0.30 | −1.43 | 0.1548 |
| 50 | 111 | −1.01 | 0.57 | −1.77 | 0.0793 |
| 52 | 281 | 0.20 | 0.22 | 0.88 | 0.3786 |
| 54 | 177 | −1.10 | 0.35 | −3.12 | 0.0021 |
| 56 | 121 | 0.81 | 0.38 | 2.11 | 0.0369 |
| 59 | 218 | 0.33 | 0.29 | 1.16 | 0.2482 |
| 61 | 135 | 0.73 | 0.37 | 1.96 | 0.0521 |

### 1.5.3 Linear mixed model results per Subject

Formula: Duration ~ LocomotorActivityMean + (1 | Day)
 Data: CallParameters[CallParameters$Subject == i,]

| Subject | N | Estimate | Std. error | T value | Pr(>|t|) |
|---------|-----|----------|-----------|---------|---------|
| 1 | 1843 | −0.83 | 0.14 | −6.06 | <0.0001 |
| 2 | 1412 | −0.90 | 0.12 | −7.57 | <0.0001 |
| 3 | 1092 | 0.06 | 0.15 | 0.37 | 0.7114 |
| 4 | 800 | −0.31 | 0.18 | −1.74 | 0.0817 |
| 5 | 1688 | −0.90 | 0.21 | −4.17 | <0.0001 |
| 6 | 1253 | −1.11 | 0.26 | −4.23 | <0.0001 |
| 7 | 1521 | −0.34 | 0.20 | −1.72 | 0.0860 |

## 1.6 Linear regression models to predict contact call Wiener entropy from locomotor activity. Results are associated with *Figure 3B*

### 1.6.1 Population-level linear mixed model results

Formula: WienerEntropyMean ~LocomotorActivityMean + (LocomotorActivityMean | Day/Subject)

Data: CallParameters
Random effects:

| Groups | Name | Variance | Std.dev. | Corr |
|---|---|---|---|---|
| Subject:Day | (Intercept) | 1.574388 | 1.25475 | |
| | LocomotorActivityMean | 1.453426 | 1.20558 | −0.51 |
| Day | (Intercept) | 1.016436 | 1.00818 | |
| | LocomotorActivityMean | 0.002195 | 0.04685 | −1.00 |
| Residual | | 1.566448 | 1.25158 | |

Number of obs: 9606, groups: Subject:Day, 188; Day, 38
Fixed effects:

| | Estimate | Std. error | Df | T value | Pr(>\|t\|) |
|---|---|---|---|---|---|
| (Intercept) | −2.5052 | 0.1914 | 37.1127 | −13.085 | 1.78e-15 |
| LocomotorActivityMean | 0.9295 | 0.1211 | 130.9984 | 7.675 | 3.36e-12 |

## 1.6.2 Linear mixed model results per Day

Formula: WienerEntropyMean ~LocomotorActivityMean + (1 | Subject)
Data: CallParameters[CallParameters$Day == i,]

| Day | N | Estimate | Std. error | T value | Pr(>\|t\|) |
|---|---|---|---|---|---|
| 1 | 471 | 0.76 | 0.31 | 2.47 | 0.0137 |
| 2 | 369 | 1.40 | 0.24 | 5.78 | <0.0001 |
| 3 | 345 | 0.68 | 0.35 | 1.94 | 0.0538 |
| 4 | 324 | 0.76 | 0.32 | 2.34 | 0.0198 |
| 5 | 353 | 0.60 | 0.32 | 1.86 | 0.0642 |
| 6 | 403 | 0.40 | 0.17 | 2.29 | 0.0228 |
| 7 | 336 | 0.82 | 0.26 | 3.16 | 0.0017 |
| 8 | 352 | 0.90 | 0.32 | 2.83 | 0.0050 |
| 9 | 379 | 0.55 | 0.34 | 1.63 | 0.1046 |
| 10 | 258 | 1.18 | 0.32 | 3.66 | 0.0003 |
| 11 | 297 | 1.17 | 0.31 | 3.84 | 0.0001 |
| 12 | 335 | 1.40 | 0.39 | 3.57 | 0.0004 |
| 13 | 348 | 1.53 | 0.28 | 5.48 | <0.0001 |
| 14 | 250 | −0.15 | 0.34 | −0.43 | 0.6656 |
| 15 | 365 | 0.66 | 0.33 | 2.02 | 0.0445 |
| 16 | 113 | 4.62 | 1.23 | 3.74 | 0.0003 |
| 17 | 241 | 0.62 | 0.46 | 1.34 | 0.1818 |
| 19 | 200 | 1.68 | 0.44 | 3.83 | 0.0002 |
| 21 | 195 | 1.72 | 0.45 | 3.83 | 0.0002 |
| 24 | 217 | 2.94 | 0.46 | 6.36 | <0.0001 |
| 26 | 255 | 1.28 | 0.50 | 2.59 | 0.0102 |
| 28 | 271 | −0.64 | 0.59 | −1.08 | 0.2828 |
| 31 | 323 | −0.75 | 0.56 | −1.33 | 0.1846 |
| 33 | 312 | 0.86 | 0.63 | 1.37 | 0.1730 |
| 35 | 222 | 1.12 | 0.59 | 1.91 | 0.0579 |
| 38 | 251 | 1.40 | 0.72 | 1.95 | 0.0521 |

| Day | N | Estimate | Std. error | T value | Pr(>|t|) |
|---|---|---|---|---|---|
| 40 | 121 | −0.55 | 1.15 | −0.48 | 0.6334 |
| 42 | 269 | 0.43 | 0.37 | 1.16 | 0.2465 |
| 45 | 259 | 1.09 | 0.77 | 1.42 | 0.1562 |
| 50 | 111 | 1.89 | 1.62 | 1.17 | 0.2464 |
| 52 | 281 | 0.64 | 0.49 | 1.30 | 0.1951 |
| 54 | 177 | −0.44 | 0.47 | −0.94 | 0.3482 |
| 56 | 121 | 1.05 | 0.80 | 1.32 | 0.1892 |
| 59 | 218 | 0.67 | 0.60 | 1.12 | 0.2650 |
| 61 | 135 | 0.29 | 0.68 | 0.43 | 0.6658 |

### 1.6.3 Linear mixed model results per Subject

Formula: WienerEntropyMean ~LocomotorActivityMean + (1 | Day)
   Data: CallParameters[CallParameters$Subject == i,]

| Subject | N | Estimate | Std. error | T value | Pr(>|t|) |
|---|---|---|---|---|---|
| 1 | 1843 | 1.25 | 0.19 | 6.46 | <0.0001 |
| 2 | 1412 | 1.33 | 0.21 | 6.48 | <0.0001 |
| 3 | 1092 | 0.13 | 0.14 | 0.95 | 0.3431 |
| 4 | 800 | 0.01 | 0.15 | 0.06 | 0.9494 |
| 5 | 1688 | 1.71 | 0.17 | 9.87 | <0.0001 |
| 6 | 1251 | 0.83 | 0.15 | 5.43 | <0.0001 |
| 7 | 1520 | 0.13 | 0.16 | 0.82 | 0.4142 |

## 1.7 Linear regression models to predict locomotor activity during contact calls from contact call type. Results are associated with *Figure 3C*

### 1.7.1 Population-level linear mixed model results

Formula: LocomotorActivityMean ~ CallType_Cry0Phee1 + (CallType_Cry0Phee1 | Day/Subject)
   Data: CallParameters
   Random effects:

| Groups | Name | Variance | Std.dev. | Corr |
|---|---|---|---|---|
| Subject:Day | (Intercept) | 0.007584 | 0.08709 | |
| | CallType_Cry0Phee1 | 0.009785 | 0.09892 | −0.98 |
| Day | (Intercept) | 0.001635 | 0.04044 | |
| | CallType_Cry0Phee1 | 0.002673 | 0.0517 | −0.99 |
| Residual | | 0.033707 | 0.18359 | |

Number of obs: 9609, groups: Subject:Day, 188; Day, 38
Fixed effects:

| | Estimate | Std. error | Df | T value | Pr(>|t|) |
|---|---|---|---|---|---|
| (Intercept) | 0.13009 | 0.01205 | 17.83393 | 10.799 | 2.98e-09 |

*continued*

|  | Estimate | Std. error | Df | T value | Pr(>\|t\|) |
|---|---|---|---|---|---|
| CallType_Cry0Phee1 | −0.07127 | 0.01373 | 19.95781 | −5.191 | 4.47e-05 |

## 1.7.2 Linear mixed model results per Day
Formula: LocomotorActivityMean ~ CallType_Cry0Phee1 + (1 | Subject)
   Data: CallParameters[CallParameters$Day == i,]

| Day | N | Estimate | Std. error | T value | Pr(>\|t\|) |
|---|---|---|---|---|---|
| 1 | 471 | −0.08 | 0.03 | −3.00 | 0.0029 |
| 2 | 369 | −0.22 | 0.04 | −6.18 | <0.0001 |
| 3 | 345 | −0.04 | 0.03 | −1.42 | 0.1568 |
| 4 | 324 | −0.06 | 0.03 | −2.10 | 0.0369 |
| 5 | 353 | −0.05 | 0.02 | −2.16 | 0.0327 |
| 6 | 403 | −0.08 | 0.03 | −2.21 | 0.0284 |
| 7 | 336 | −0.10 | 0.03 | −3.25 | 0.0014 |
| 8 | 352 | −0.06 | 0.02 | −2.51 | 0.0132 |
| 9 | 379 | −0.05 | 0.02 | −2.12 | 0.0380 |
| 10 | 258 | −0.16 | 0.03 | −4.70 | <0.0001 |
| 11 | 297 | −0.13 | 0.03 | −4.97 | <0.0001 |
| 12 | 336 | −0.07 | 0.02 | −3.22 | 0.0032 |
| 13 | 348 | −0.17 | 0.03 | −6.28 | <0.0001 |
| 14 | 250 | 0.01 | 0.02 | 0.40 | 0.6867 |
| 15 | 365 | −0.10 | 0.02 | −5.03 | <0.0001 |
| 16 | 113 | −0.13 | 0.03 | −3.70 | 0.0003 |
| 17 | 241 | −0.11 | 0.03 | −3.34 | 0.0038 |
| 19 | 200 | −0.23 | 0.03 | −6.51 | <0.0001 |
| 21 | 195 | −0.32 | 0.04 | −8.01 | <0.0001 |
| 24 | 217 | −0.32 | 0.04 | −7.98 | <0.0001 |
| 26 | 255 | −0.11 | 0.04 | −2.73 | 0.0156 |
| 28 | 272 | −0.02 | 0.02 | −0.98 | 0.3512 |
| 31 | 323 | 0.00 | 0.02 | 0.12 | 0.9076 |
| 33 | 312 | −0.09 | 0.04 | −2.42 | 0.0162 |
| 35 | 222 | −0.05 | 0.07 | −0.78 | 0.4344 |
| 38 | 251 | 0.00 | 0.03 | −0.10 | 0.9180 |
| 40 | 122 | −0.03 | 0.03 | −0.91 | 0.3650 |
| 50 | 111 | 0.07 | 0.10 | 0.67 | 0.5068 |

## 1.7.3 Linear mixed model results per Subject
Formula: LocomotorActivityMean ~ CallType_Cry0Phee1 + (1 | Day)
   Data: CallParameters[CallParameters$Subject == i,]

| Subject | N | Estimate | Std. error | T value | Pr(>\|t\|) |
|---|---|---|---|---|---|
| 1 | 1843 | −0.07 | 0.01 | −7.20 | <0.0001 |
| 2 | 1412 | −0.15 | 0.02 | −9.19 | <0.0001 |

*continued*

| Subject | N | Estimate | Std. error | T value | Pr(>|t|) |
|---------|------|----------|------------|---------|----------|
| 3 | 1092 | −0.01 | 0.01 | −0.49 | 0.6280 |
| 4 | 800 | −0.02 | 0.02 | −1.00 | 0.3172 |
| 5 | 1688 | −0.13 | 0.01 | −10.97 | <0.0001 |
| 6 | 1253 | −0.10 | 0.01 | −8.48 | <0.0001 |
| 7 | 1521 | −0.04 | 0.01 | −2.96 | 0.0035 |

## 1.8 Linear regression models to predict locomotor activity during mature contact calls from age group (early vs late development). Results are associated with *Figure 3D*

### 1.8.1 Population-level linear mixed model results

Formula: LocomotorActivityMean ~ AgeGroup_Early0Late1 + (AgeGroup_Early0Late1| Day/Subject)
  Data: CallParameters[CallParameters$CallType_Cry0Phee1 == 1,]
  Random effects:

| Groups | Name | Variance | Std.dev. | Corr |
|--------|------|----------|----------|------|
| Subject:Day | (Intercept) | 7.276e-04 | 0.0269731 | |
| | AgeGroup_Early0Late1 | 1.774e-03 | 0.0421153 | 0.39 |
| Day | (Intercept) | 0.000e+ | 00 | 0.0000000 |
| | AgeGroup_Early0Late1 | 1.060e-07 | 0.0003255 | NaN |
| Residual | | 1.904e-02 | 0.1379996 | |

  Number of obs: 1846, groups: Subject:Day, 82; Day, 15
  Fixed effects:

| | Estimate | Std. error | Df | T value | Pr(>|t|) |
|--|----------|------------|-----|---------|----------|
| (Intercept) | 0.043492 | 0.006165 | 50.194174 | 7.055 | 4.83e-09 |
| AgeGroup_Early0Late1 | 0.042408 | 0.01392 | 23.693856 | 3.047 | 0.00561 |

### 1.8.2 Linear mixed model results per Subject

Formula: LocomotorActivityMean ~ AgeGroup_Early0Late1 + (1|Day)
  Data: CallParameters[CallParameters$CallType_Cry0Phee1 == 1 and CallParameters$Subject == i,])

| Subject | N | Estimate | Std. error | T value | Pr(>|t|) |
|---------|-----|----------|------------|---------|----------|
| 1 | 295 | 0.005 | 0.020 | 0.27 | 0.7998 |
| 2 | 339 | −0.021 | 0.027 | −0.79 | 0.4474 |
| 3 | 274 | 0.016 | 0.024 | 0.65 | 0.5170 |
| 4 | 201 | 0.003 | 0.027 | 0.11 | 0.9109 |
| 5 | 272 | 0.186 | 0.042 | 4.41 | 0.0009 |
| 6 | 262 | 0.046 | 0.025 | 1.84 | 0.0789 |
| 7 | 203 | 0.063 | 0.035 | 1.79 | 0.0932 |

## 1.9 Linear mixed models to predict heart rate percentile during contact calls from contact call type. Results are associated with *Figure 4A*

### 1.9.1 Population-level linear mixed model results

Formula: HeartRatePercentileMean ~CallType_Cry0Phee1 + (CallType_Cry0Phee1 | Day/Subject)

Data: CallParameters
Random effects:

| Groups | Name | Variance | Std.dev. | Corr |
|---|---|---|---|---|
| Subject:Day | (Intercept) | 72.37 | 8.507 | |
| | CallType_Cry0Phee1 | 96.64 | 9.831 | −0.53 |
| Day | (Intercept) | 0.00 | 0.000 | |
| | CallType_Cry0Phee1 | 10.21 | 3.195 | NaN |
| Residual | | 611.88 | 24.736 | |

Number of obs: 6215, groups: Subject:Day, 149; Day, 37
Fixed effects:

| | Estimate | Std. error | Df | T value | Pr(>\|t\|) |
|---|---|---|---|---|---|
| (Intercept) | 44.630 | 1.142 | 58.095 | 39.078 | <2e-16 |
| CallType_Cry0Phee1 | 3.376 | 1.453 | 27.999 | 2.324 | 0.0276 |

### 1.9.2 Linear mixed model results per Day

Formula: HeartRatePercentileMean ~CallType_Cry0Phee1 + (1 | Subject)
Data: CallParameters[CallParameters$Day == i,]

| Day | N | Estimate | Std. error | T value | Pr(>\|t\|) |
|---|---|---|---|---|---|
| 1 | 48 | −9.37 | 6.39 | −1.47 | 0.1495 |
| 2 | 252 | −9.85 | 4.22 | −2.33 | 0.0204 |
| 3 | 120 | 15.52 | 4.70 | 3.30 | 0.0013 |
| 4 | 239 | 5.42 | 3.10 | 1.75 | 0.0817 |
| 5 | 123 | 4.54 | 4.17 | 1.09 | 0.2781 |
| 6 | 213 | 2.95 | 3.58 | 0.82 | 0.4110 |
| 7 | 187 | 5.65 | 5.58 | 1.01 | 0.3124 |
| 8 | 206 | 1.86 | 4.06 | 0.46 | 0.6499 |
| 9 | 198 | −2.16 | 4.69 | −0.46 | 0.6494 |
| 10 | 189 | −1.67 | 4.79 | −0.35 | 0.7280 |
| 11 | 148 | −0.39 | 5.45 | −0.07 | 0.9425 |
| 12 | 244 | −9.02 | 4.05 | −2.22 | 0.0272 |
| 13 | 220 | 3.91 | 4.39 | 0.89 | 0.3743 |
| 14 | 226 | 14.57 | 5.10 | 2.86 | 0.0094 |
| 15 | 285 | 10.22 | 3.38 | 3.03 | 0.0031 |
| 17 | 157 | 3.45 | 7.54 | 0.46 | 0.6549 |
| 19 | 41 | 5.78 | 9.74 | 0.59 | 0.5815 |
| 21 | 76 | −1.05 | 6.91 | −0.15 | 0.8793 |
| 24 | 198 | 6.24 | 7.25 | 0.86 | 0.3922 |

*continued*

| Day | N | Estimate | Std. error | T value | Pr(>\|t\|) |
|---|---|---|---|---|---|
| 26 | 204 | 6.74 | 5.25 | 1.28 | 0.2556 |
| 28 | 213 | 36.94 | 9.67 | 3.82 | 0.0003 |
| 31 | 231 | 10.27 | 5.03 | 2.04 | 0.0438 |
| 33 | 277 | −2.37 | 8.83 | −0.27 | 0.7882 |
| 35 | 214 | 9.03 | 13.51 | 0.67 | 0.5044 |
| 38 | 225 | 1.66 | 5.01 | 0.33 | 0.7423 |
| 40 | 112 | −2.61 | 6.57 | −0.40 | 0.6913 |

### 1.9.3 Linear mixed model results per Subject

Formula: HeartRatePercentileMean ~CallType_Cry0Phee1 + (1 | Day)
  Data: CallParameters[CallParameters$Subject == i,]

| Subject | N | Estimate | Std. error | T value | Pr(>\|t\|) |
|---|---|---|---|---|---|
| 1 | 677 | 1.09 | 2.83 | 0.39 | 0.7003 |
| 2 | 454 | −4.86 | 4.23 | −1.15 | 0.2528 |
| 3 | 1011 | 6.42 | 2.53 | 2.54 | 0.0114 |
| 4 | 731 | 3.40 | 3.02 | 1.12 | 0.2618 |
| 5 | 1261 | −0.88 | 1.87 | −0.47 | 0.6393 |
| 6 | 935 | 10.80 | 2.55 | 4.23 | <0.0001 |
| 7 | 1146 | 7.68 | 2.20 | 3.49 | 0.0006 |

## 1.10 Linear regression models to predict heart rate percentile during mature contact calls from age group (early vs late development). Results are associated with *Figure 4B*

### 1.10.1 Population-level linear mixed model results

Formula: HeartRatePercentileMean ~AgeGroup_Early0Late1 + (AgeGroup_Early0Late1 | Day/Subject)
  Data: CallParameters[CallParameters$CallType_Cry0Phee1 == 1,]
  Random effects:

| Groups | Name | Variance | Std.dev. | Corr |
|---|---|---|---|---|
| Subject:Day | (Intercept) | 9.717e + 01 | 9.857339 | |
| | AgeGroup_Early0Late1 | 3.598e + 02 | 18.967551 | −0.82 |
| Day | (Intercept) | 2.077e-06 | 0.001441 | |
| | AgeGroup_Early0Late1 | 9.461e-06 | 0.003076 | −0.88 |
| Residual | | 5.561e + 02 | 23.580851 | |

Number of obs: 1270, groups: Subject:Day, 58; Day, 15
Fixed effects:

| | Estimate | Std. error | Df | T value | Pr(>\|t\|) |
|---|---|---|---|---|---|
| (Intercept) | 48.13 | 2.074 | 28.722 | 23.205 | <2e-16 |
| AgeGroup_Early0Late1 | 6.277 | 3.478 | 36.196 | 1.805 | 0.0795 |

### 1.10.2 Linear model results per Subject

Formula: HeartRatePercentileMean ~AgeGroup_Early0Late1 + (1|Day)
    Data: CallParameters[CallParameters$CallType_Cry0Phee1 == 1 and CallParameters$Subject == i,])

| Subject | N | Estimate | Std. error | T value | Pr(>|t|) |
|---|---|---|---|---|---|
| 1 | 110 | 9.46 | 14.45 | 0.65 | 0.5524 |
| 2 | 69 | −13.20 | 13.26 | −1.00 | 0.4793 |
| 3 | 268 | −14.30 | 5.35 | −2.67 | 0.0440 |
| 4 | 184 | 10.78 | 6.82 | 1.58 | 0.1521 |
| 5 | 211 | 24.63 | 10.99 | 2.24 | 0.0521 |
| 6 | 245 | 10.84 | 6.56 | 1.65 | 0.1424 |
| 7 | 183 | 2.03 | 10.53 | 0.19 | 0.8526 |

## 1.11 Linear regression models to predict heart rate percentile during locomotor events from age group (early vs late development). Results are associated with *Figure 4D*

### 1.11.1 Population-level linear mixed model results

Formula: HeartRatePercentileMean ~AgeGroup_Early0Late1 + (AgeGroup_Early0Late1| Day/Subject)
    Data: LocomotorParameters
    Random effects:

| Groups | Name | Variance | Std.dev. | Corr |
|---|---|---|---|---|
| Subject:Day | (Intercept) | 89.14 | 9.441 | |
| | AgeGroup_Early0Late1 | 369.73 | 19.228 | −0.96 |
| Day | (Intercept) | 61.31 | 7.830 | |
| | AgeGroup_Early0Late1 | 61.36 | 7.833 | −1.00 |
| Residual | | 669.67 | 25.878 | |

Number of obs: 1318, groups: Subject:Day, 58; Day, 15
Fixed effects:

| | Estimate | Std. error | Df | T value | Pr(>|t|) |
|---|---|---|---|---|---|
| (Intercept) | 44.717 | 3.161 | 8.974 | 14.15 | 1.93e-07 |
| AgeGroup_Early0Late1 | 8.116 | 4.039 | 20.251 | 2.01 | 0.058 |

### 1.11.2 Linear mixed model results per Subject

Formula: HeartRatePercentileMean ~AgeGroup_Early0Late1 + (1|Day)
    Data: LocomotorParameters[LocomotorParameters$Subject == i,])

| Subject | N | Estimate | Std. error | T value | Pr(>|t|) |
|---|---|---|---|---|---|
| 1 | 106 | −1.52 | 6.40 | −0.24 | 0.8126 |
| 2 | 38 | −20.34 | 28.04 | −0.73 | 0.5537 |
| 3 | 254 | −0.50 | 8.84 | −0.06 | 0.9569 |

| Subject | N | Estimate | Std. error | T value | Pr(>\|t\|) |
|---|---|---|---|---|---|
| 4 | 187 | −1.24 | 10.45 | −0.12 | 0.9092 |
| 5 | 235 | 17.08 | 7.80 | 2.19 | 0.0574 |
| 6 | 252 | 10.14 | 3.51 | 2.89 | 0.0042 |
| 7 | 246 | 20.54 | 6.46 | 3.18 | 0.0137 |

*continued*

## 1.12 Linear regression models to predict heart rate percentile from locomotor activity during mature contact calls. Results are associated with *Figure 5B*

### 1.12.1 Population-level linear mixed model results

Formula: HeartRatePercentileMean ~LocomotorActivityMean + (LocomotorActivityMean | Day/Subject)

Data: CallParameters[CallParameters$CallType_Cry0Phee1 == 1,]

Random effects:

| Groups | Name | Variance | Std.dev. | Corr |
|---|---|---|---|---|
| Subject:Day | (Intercept) | 79.08 | 8.893 | |
| | LocomotorActivityMean | 636.31 | 25.225 | −0.17 |
| Day | (Intercept) | 11.64 | 3.412 | |
| | LocomotorActivityMean | 27.74 | 5.267 | −1.00 |
| Residual | | 581.75 | 24.120 | |

Number of obs: 3966, groups: Subject:Day, 144; Day, 37

Fixed effects:

| | Estimate | Std. error | Df | T value | Pr(>\|t\|) |
|---|---|---|---|---|---|
| (Intercept) | 47.585 | 1.093 | 35.736 | 43.538 | <2e-16 |
| LocomotorActivityMean | 9.666 | 4.046 | 49.36 | 2.389 | 0.0208 |

## 1.12.2 Population-level linear mixed model results, data limited to postnatal days 1–30

Formula: HeartRatePercentileMean ~LocomotorActivityMean + (LocomotorActivityMean | Day/Subject)

Data: CallParameters[CallParameters$CallType_Cry0Phee1 == 1 and CallParameters$Day <= 30,]

Random effects:

| Groups | Name | Variance | Std.dev. | Corr |
|---|---|---|---|---|
| Subject:Day | (Intercept) | 96.5718 | 9.8271 | |
| | LocomotorActivityMean | 1097.4655 | 33.1280 | −0.64 |
| Day | (Intercept) | 0.1134 | 0.3368 | |
| | LocomotorActivityMean | 1.1894 | 1.0906 | 1.00 |
| Residual | | 592.4485 | 24.3403 | |

Number of obs: 1705, groups: Subject:Day, 86; Day, 22

Fixed effects:

*continued*

|  | Estimate | Std. error | Df | T value | Pr(>\|t\|) |
|---|---|---|---|---|---|
|  | Estimate | Std. error | Df | T value | Pr(>\|t\|) |
| (Intercept) | 47.163 | 1.335 | 44.793 | 35.333 | <2e-16 |
| LocomotorActivityMean | 16.317 | 6.603 | 31.446 | 2.471 | 0.0191 |

### 1.12.3 Population-level linear mixed model results, data limited to postnatal days 31–61

Formula: HeartRatePercentileMean ~LocomotorActivityMean + (LocomotorActivityMean | Day/Subject)

 Data: CallParameters[CallParameters$CallType_Cry0Phee1 == 1 and CallParameters$Day > 30,]

 Random effects:

| Groups | Name | Variance | Std.dev. | Corr |
|---|---|---|---|---|
| Subject:Day | (Intercept) | 60.96 | 7.808 |  |
|  | LocomotorActivityMean | 421.40 | 20.528 | 0.41 |
| Day | (Intercept) | 29.96 | 5.473 |  |
|  | LocomotorActivityMean | 74.54 | 8.634 | −1.00 |
| Residual |  | 572.1 | 23.919 |  |

 Number of obs: 2261, groups: Subject:Day, 58; Day, 15

 Fixed effects:

|  | Estimate | Std. error | Df | T value | Pr(>\|t\|) |
|---|---|---|---|---|---|
| (Intercept) | 48.216 | 1.911 | 14.220 | 25.235 | 3.27e-13 |
| LocomotorActivityMean | 2.627 | 5.242 | 18.818 | 0.501 | 0.622 |

### 1.12.4 Linear mixed model results per Day

Formula: HeartRatePercentileMean ~LocomotorActivityMean + (1 | Subject)

 Data: CallParameters[CallParameters$CallType_Cry0Phee1 == 1 and CallParameters$Day == i,]

| Day | N | Estimate | Std. error | T value | Pr(>\|t\|) |
|---|---|---|---|---|---|
| 2 | 56 | 52.99 | 29.90 | 1.77 | 0.0820 |
| 3 | 46 | −9.96 | 19.25 | −0.52 | 0.6076 |
| 4 | 70 | 13.73 | 18.41 | 0.75 | 0.4584 |
| 5 | 37 | 4.96 | 25.53 | 0.19 | 0.8471 |
| 6 | 43 | −96.73 | 36.48 | −2.65 | 0.0115 |
| 7 | 25 | 8.04 | 53.55 | 0.15 | 0.8819 |
| 8 | 62 | −24.79 | 19.54 | −1.27 | 0.2096 |
| 9 | 54 | 62.74 | 22.24 | 2.82 | 0.0068 |
| 10 | 66 | 49.74 | 31.19 | 1.59 | 0.1161 |
| 11 | 40 | 87.39 | 34.41 | 2.54 | 0.0153 |
| 12 | 115 | −5.61 | 16.32 | −0.34 | 0.7317 |
| 13 | 70 | −19.29 | 32.42 | −0.59 | 0.5540 |
| 14 | 146 | 13.54 | 13.36 | 1.01 | 0.3125 |

*continued*

| Day | N | Estimate | Std. error | T value | Pr(>|t|) |
|-----|-----|----------|------------|---------|----------|
| 15 | 146 | 40.18 | 17.61 | 2.28 | 0.0239 |
| 17 | 110 | 18.66 | 22.67 | 0.82 | 0.4122 |
| 19 | 15 | −23.71 | 26.24 | −0.90 | 0.3827 |
| 21 | 59 | 9.76 | 25.17 | 0.39 | 0.6998 |
| 24 | 174 | 6.51 | 16.62 | 0.39 | 0.6957 |
| 26 | 168 | 39.18 | 12.93 | 3.03 | 0.0028 |
| 28 | 181 | 15.80 | 15.00 | 1.05 | 0.2938 |
| 31 | 159 | −10.56 | 13.82 | −0.76 | 0.4461 |
| 33 | 270 | −8.99 | 10.22 | −0.88 | 0.3799 |
| 35 | 211 | −3.98 | 11.09 | −0.36 | 0.7202 |
| 38 | 177 | 14.31 | 12.22 | 1.17 | 0.2433 |
| 40 | 93 | 51.52 | 29.90 | 1.72 | 0.0882 |
| 42 | 262 | 11.59 | 8.97 | 1.29 | 0.1975 |
| 45 | 181 | 44.64 | 12.13 | 3.68 | 0.0003 |
| 52 | 241 | −1.05 | 10.90 | −0.10 | 0.9234 |
| 54 | 175 | −48.91 | 15.80 | −3.10 | 0.0023 |
| 56 | 119 | 25.30 | 17.63 | 1.43 | 0.1540 |
| 59 | 121 | −6.29 | 11.51 | −0.55 | 0.5855 |
| 61 | 124 | −23.32 | 14.14 | −1.65 | 0.1017 |

## 1.12.5 Linear mixed model results per Subject

Formula: HeartRatePercentileMean ~LocomotorActivityMean + (1 | Day)
    Data: CallParameters[CallParameters$CallType_Cry0Phee1 == 1 and CallParameters$Subject == i,]

| Subject | N | Estimate | Std. error | T value | Pr(>|t|) |
|---------|-----|----------|------------|---------|----------|
| 1 | 447 | 21.03 | 8.61 | 2.44 | 0.0150 |
| 2 | 351 | −3.12 | 10.74 | −0.29 | 0.7713 |
| 3 | 826 | −3.56 | 5.00 | −0.71 | 0.4770 |
| 4 | 607 | 0.71 | 5.63 | 0.13 | 0.9002 |
| 5 | 775 | 31.31 | 6.97 | 4.49 | <0.0001 |
| 6 | 566 | −9.67 | 9.56 | −1.01 | 0.3120 |
| 7 | 394 | 18.93 | 10.32 | 1.83 | 0.0674 |

