## [Decision Letter]

Thank you for submitting your article "Energetic costs and locomotor constraints on vocal development" for consideration by *eLife*. Your article has been reviewed by three peer reviewers, one of whom as a guest Reviewing Editor, and the evaluation has been overseen by Ronald Calabrese as the Senior Editor. The following individuals involved in review of your submission have agreed to reveal their identity: Franz Goller (Reviewer #3).

The reviewers have discussed the reviews with one another and the Reviewing Editor has drafted this decision to help you prepare a revised submission.

Summary:

Gustison and colleagues recorded and analyzed the development of coordination between vocal and locomotor behaviors in the marmoset monkey. Results show that mature vocalizations develop earlier than locomotion. Further, coordination between vocalization and locomotion develop gradually. Younger infants produce mature-sounding vocalizations only while not moving. Older infants, on the other hand, coordinated their vocalization with movements. Recording heart rate suggest that energetics constraints could potentially explain the late development of coordinated behavior across modalities. However, the reviewers found some substantive concerns, as elaborated below.

Essential revisions:

There are two very major concerns that should be fully addressed before resubmission can be considered:

First, all reviewers concluded (during discussion) that data do not support the major claims authors make about energetic constraints on the development of vocal-locomotor coordination. Therefore, authors should drop out heart rate as a measure of metabolic cost of vocalization. Instead, the revised manuscript should be focused on describing the developmental trajectories with more details, and compare them to relevant literature in other species (which should be presented in Introduction).

Second, authors used sessions (and sometimes even calls) as independent measurements to make statistical inferences. In the revised version, statistics should be redone using each animal as a statistic (i.e., with df = number of monkeys -1). It is essential to test if the phenomenon of different maturation rate can hold using simple paired comparisons. Further, performing the statistics on the fitted spline curves makes it hard to assess if conclusions are sufficiently supported. The dynamic analysis of calls/locomotion coordination should also be supported with similar statistics.

*Reviewer #1:*

This is an important longitudinal study that compare maturation rate of vocalization to movement and postures. Authors discoveries, if correct, are of general theoretical and practical interest. Of particular interest is the dynamic analysis of movements and calls coordination, showing reversal of patterns over development. Before this manuscript can be published, several concerns about the statistical approach and about conclusions should be addressed:

A) Statistical approach:

1) Using each call as independent statistic is problematic, and (to my knowledge) not an acceptable practice in acoustic analysis. At least for the most critical tests, the n must be the number of monkeys studied, namely 7.

2) Authors should be more careful with their statistical modeling, and take a simple, direct approach whenever possible. In particular, when comparing time courses of calls and movement maturity indexes, author should simply test if those time courses are different. That is, the null hypothesis should be that those are the same. Here all we need is a pairwise comparison within each subject: is the zero crossing the same or not? Such pairwise comparison with n=7 using paired Wilcoxon, or even paired t or just binomial test would be (one can safely reject the null hypothesis (with p=1/2^7) and conclude that the maturation time course of vocalization is indeed faster.

3) The way authors did the test is inappropriate in several ways. First, raising 3 distinct hypotheses (H1= more, H2=same, H3=less) is not necessary, it suffice to look for a significant difference from null. Second, and most critical – where do 124 degrees of freedom comes from? This looks like an obvious pseudo-replication, which resulted with illegitimate power of W = 3936 (!) and an erroneously huge significance (p < 0.0001). Such p values are rarely correct in biology, certainly not with the sample size of the current study.

4) Similarly, the analysis of movement vs. Wiener entropy should be done with each monkey as a statistic, and again, paired comparisons should be done within each animal.

5) The only place where I find it reasonable to use calls as statistics is in the dynamic analysis shown in Figures 4 and 5. In such a fine grain developmental analysis, and after the major effects where established, it makes sense to look at time courses of call movements interactions, considering each event as statistically independent.

B) Conclusions:

This is an important, but still a correlational study – we don't know cause and effect. For example, the statement "infants must overcome the energetic costs of coordinating their vocalizations with body movement" is not fully supported by the data. For example, the marmoset is likely to vocalize while already agitated and this effect might be age related. Further, one might suspect that vocalization might be used to communicate high energy/agitation state.

*Reviewer #2:*

In this very interesting and well-written manuscript Gustison et al. address the maturation of two motor systems (vocal and locomotor) in marmosets and test a string of clearly defined hypotheses. The authors quantify vocal, postural and locomotory behaviors over complete postnatal development and the resulting datasets are impressive. They use these observations to compute maturity indices for the three behavioral classes, allowing comparison of maturity onsets for the systems separately. The authors convincingly show that young infants only produce adult like calls when not moving and older ones can move while calling. The paper includes several novelties: i) concurrent analyses of both the developmental timescale as well as the real-time (seconds) timescale to investigate energetic tradeoffs during vocalizations. ii) which allows to them study coupling for the first time. The authors show that this coupling exists at the second scale suggesting higher energy investment in calling behavior is required for producing mature calls. The data is well presented with clear figures.

Major comments:

1) The data is used to support the hypothesis that mature contact call production requires an energy investment that increases with age. In the light of energy expenditure during vocalizations this can be explained if the calls get louder. Increasing respiratory effort and thus lung pressure increases predominantly sound amplitude in laryngeally produced sounds. Fundamental frequency only changes slightly with pressure. Wiener entropy as a measure for noisiness is not regularly used outside of the birdsong field, but will increase most likely in chaotic regimes when pressures are very high.

It would be rather straightforward for the authors to quantify the source levels of the vocalizations. The low-frequency cry is not directional but the wave-number can be used to coarsely estimate the directionality of the phee call. (ka = 2pi/λ * a (, where λ is wavelength (v/f) and a is emitter size radius – guessing 2 cm for a marmoset. with k=2pi/(340/7000) and a = 0.02) ka=2.5, which is above 1. Thus the phee call is directional, which makes it more complicated to compute source levels without microphone array. However luckily the microphone was located in the far-field 90 cm above the cage directing down, making vocalization's aimed in the horizontal plane directly comparable.

2) The authors list four empirical data papers in the last paragraph of the discussion that investigated human infant vocal-locomotor coordination. At least Berger et al. and Abney et al. present similar datasets of concurrent posture and vocalization (not at the second timescale). These papers should be acknowledged and placed in perspective up front in the Introduction.

3) The locomotor behavior was basically scored as difference images. Did the authors observe any structural changes in locomotory patterns, such as certain accompanying displays (vocal postures) that matured over development?

*Reviewer #3:*

General comments:

This manuscript describes a research effort, in which development of locomotor and vocal skills as well as, importantly, their coordination have been studied quantitatively and have been related to energetic aspects. This is a very interesting study that tries to fill a gap in our understanding of the ontogeny of motor system coordination and its energetic background. The presented data partly meet the goals set out in the Introduction (questions 1-3), but especially for questions 2 and 3 fall somewhat short and require clarification.

1) The various sets of hypotheses listed in the results present sets that leave out potential alternatives. Specific suggestions are mentioned below.

2) The assessment of the energetic cost of call production is problematic at multiple levels, from the premises to the estimate and the interpretation. Specifically, the following general points should be considered:

2.1) Several assumptions had to be made to convert heart rate to metabolic activity. We know from critical assessments of the technique that careful calibrations are needed to achieve reliable estimates. That means that the applied conversion factors may not be valid across development within an individual and certainly are problematic when applied across individuals. Whereas this is a general difficulty with the technique that cannot be solved in this study, it should be acknowledged and should lead to more careful conclusions.

2.2) It is not clear to me how baseline levels of heart rate have been determined for call production episodes with and without locomotor activity. Because heart rate changes are very dynamic, this is one of the critical methodological issues, which we face. Are observed changes due to actual changes or differences in baseline levels?

2.3) Rapid changes in heart rate can be caused by motivational and other modulatory factors that have no direct link to energy expenditure of a behavior. This is a major shortcoming of using heart rate to assess the metabolic cost for short-duration behaviors.

2.4) It is not clear to me how the cost of locomotion, which is expected to be several-fold greater than that for vocalization has been disentangled to conclude that production of phee calls imposes a high metabolic cost.

3) The presentation of the results is interwoven by methods (somewhat necessitated by the fact that the method section is at the end) and lots of introductory material and posing of hypotheses. The actual results are then described very briefly, and statistical as well as descriptive presentation is quickly reduced to very derived spline curves. At least to me this generates the feeling that I do not know the results sufficiently to assess whether or not the later conclusions are supported sufficiently.

Specific comments:

Abstract: Is the intended meaning that locomotion uses resources that would otherwise be available for call production? I cannot see that such an energetic bottleneck exists. If a constraint exists, it is more likely that locomotion and call production require either muscle efforts that are difficult to generate simultaneously or some other coordination-related trade-off. The absolute energy level is very unlikely to be the bottle neck here.

Introduction paragraph one: Of course, there are always energetic costs to vocal production. How much energy is required may vary, but as discussed above it is hardly so high that energy levels prohibit vocalization.

Subsection “The vocal system matures before postural and locomotor systems” paragraph two: This description of hypotheses is almost "trivial" and thus does not need to be so elaborate.

Subsection “Mature contact call production and locomotor activity become increasingly coordinated during development”: These hypotheses are based on the same notion that absolute energy levels at the time of vocalization could dictate whether or not calls can be produced. I do not see that such an energy shortage could exist. It is more likely (as alluded to above) that other factors dictate whether or not a call can be produced (e.g., can a particular respiratory muscle generate sufficient force if it also is engaged in locomotion, etc.). Another matter is whether or not an animal engages in a particular behavior because of general energetic condition. These conceptual issues are at the heart of this paper, and, in my opinion, the current reasoning is not realistic in regard to the physiological understanding of metabolic cost and movement.

Subsection “Mature contact call production and locomotor activity become increasingly coordinated during development”: This discussion would read very differently if the constraint were seen more like discussed above.

Final sentence of subsection “Mature contact call production and locomotor activity become increasingly coordinated during development”: This statement may have to be revised if calibrations at different ages were available.

Materials and methods: It is not clear how baseline heart rate was determined. Because metabolic cost of locomotion is not restricted in time to the actual duration of movement, but elevated metabolism continues after the movement, the timing of call production relative to locomotion is very important. This issue is very critical, if the costs of locomotion and vocalization are to be disentangled. It is also not clear how z-scores were calculated to make data comparable over developmental time.

[Editors' note: comments from the first round of re-review follow.]

Thank you for resubmitting your work entitled "Coordination of vocal and locomotor behavior emerges in development as a function of arousal state" for further consideration at *eLife*. The manuscript has been improved but there are some remaining issues that need to be addressed before acceptance, as outlined below.

In particular:

1) Statistics need improvement. Please provide cross validation by doing shuffle statistics (shuffling subjects across groups). Also, if you choose to continue with the GLM please provide detailed statistical results including within and between subjects results.

2) As reviewer 3 noted, the problem of disentangling metabolism related to locomotion from emotional influences is not fully resolved by using the term 'arousal'. A weaker interpretation of the results could help (in addition to clarifying terminology as suggested).

*Reviewer #1:*

Authors addressed my comments and the manuscript is much improved but I am still worried about the statistics. Authors now use GLM throughout but they did not specify how dependent repeated observations are treated in the model. judged by the p values, which are all lower than 0.0001, I suspect that p values are based on the dependent call events and movement events as the statistics, rather than the animal. This is a problem. I will need to see the breakdown of within and between subject results, and how p values were computed to confirm, but looking at the raw data, I am almost sure that p values are wrong. Authors should keep in mind that as opposed to R, Matlab is not a statistical software, and it is very easy to get the model wrong. One should use Matlab statistical packages carefully, particularly when extracting p values from complex models, to make sure all assumptions are correct. I still believe it would be much safer to shuffle statistics here: simply shuffle subjects across groups and extract direct p values. Simple is better than sophisticated.

*Reviewer #2:*

After carefully reviewing the response to reviewers and the new manuscript, I do not have any substantial concerns and recommend the paper for publication.

*Reviewer #3:*

The manuscript has been improved following the suggestions. The metabolic data and associated argumentation have been taken out. In its place, the heart rate data were used to assess "arousal". "Arousal" is a vague concept and is not defined very well in this manuscript. Whereas I do not see the issues as critical as in the case of metabolism, I am not clear on whether or not the problem of disentangling metabolism related to locomotion from emotional influences has been solved any better here. Perhaps I still do not understand it correctly, but the issue of baseline data has in my opinion also not been resolved satisfactorily. Namely, the interpretation of whether or not and how much heart rate increased depends on the period of baseline measurement. I am not sure I understand what a session is (letter) for which the normalized data are taken. Is it the period before and after as outlined in the manuscript? Certainly, different locomotor activity during these periods will make comparisons very difficult. Because locomotor behavior increases with maturity level, so will the associated heart rates. To what degree it is a change in arousal, can therefore only be said with confidence if the two influences on heart rate can be disentangled.

[Editors' note: comments from the second round of re-review follow.]

Thank you for resubmitting your work entitled "Coordination of vocalization and locomotion emerges in development in association with arousal state" for further consideration at *eLife*. Your revised article has been evaluated by Ronald Calabrese (Senior Editor), a Reviewing Editor, and a statistician expert. Unfortunately, there are still very serious issues with statistics that preclude publication of your manuscript in *eLife* at its present form.

Following your disagreement with the reviewers’ criticism about statistics, we sent your manuscript to a statistician expert. The expert concerns are similar to those raised earlier, but are even more substantial. We are worried about the p-values you reported, which seem inconsistent with your sample size and with the raw measures. Therefore, we cannot accept your manuscript, unless we can validate and replicate your results, according to the guidelines provided by the statistician expert.

We would like to clarify that as much as we find your study interesting and potentially valuable, unless the revised version can fully convince the expert that the stats and p-values are correct and easily replicated, we will have no choice but to reject your manuscript.

*Reviewer #4:*

Major concerns:

1) Authors use LMM which is fine. They say they have infants as random effect and that is fine too, but I wonder how could authors get with 7 monkeys p<0.0001? I looked at the tables with per-animal results. The variability is high in the coefficients per monkey, sometimes with results of opposite directions. Treating them as a sample does not yield the precision near to that claimed.

2) I cannot exclude the possibility that authors may be right, BUT when submitting such an analysis reproducible computing is a must. The statistical model itself should be specified and the entire code made available. I think that giving this information is minimal requirement, but making the code and the data available at this stage so that reproducibility could be checked is already a requirement made by leading journals and here it is crucial.

3) In particular in this case, it is important to know whether the model considered sessions as nested within monkeys or not. It will have great impact, and you cannot tell it from the information authors provided.

4) In the correlation analysis authors use all observations as if they are independent and that is wrong, but since it is used only for creation of scale it does not matter much.

5) Authors also present too many figures only with no data, only summaries: all of 4 and 5B.

6) *eLife* requires treatment of multiplicity issues; none of that was done.

[Editors' note: comments from the third round of re-review follow.]

Thank you for submitting your article "Vocal and locomotor coordination develops in association with arousal state" for consideration by *eLife*. Your article has been reviewed by one peer reviewer, and the evaluation has been overseen by a Reviewing Editor and Ronald Calabrese as the Senior Editor.

The reviewers have discussed the reviews with one another and the Reviewing Editor has drafted this decision to help you prepare a revised submission.

Summary:

We appreciate authors efforts to correct the statistical issues. However, there are still major problems with statistics that authors need to address before the manuscript can be accepted. Please carefully follow the statistician guidance, and in particular correct the models to include individual slopes, and properly adjust p values for multiplicity. Please send us correspondence if statistical significance of the major finding is no longer valid.

*Reviewer #4:*

Reviewer notes on "Vocal and Locomotor coordination…" by Gustison et al.

I thank the authors for their response to my previous questions. Their effort to make their analysis transparent and reproducible are evident in this version. I hope this version can serve as a model for reproducible research for others.

These compliments do not mean that there are no problems with their analyses,

I shall list three of them:

1) The model they used allows only random intercepts, and not random slopes.

hence the variability from one monkey to the other is not reflected in the calculations of the slope's standard deviation and p-value, which are far too small. In order to enhance replicability of their results, when the experiment will be conducted on another set of monkeys the random slopes inference is important (Then the models should be ~Day + Day/Subject)

The analyses for individuals is appropriate so it lets assess how serious this flaw is.

Sometimes it is very inappropriate: HeartRatePercentileMean~CallType_Cry0Phee1

The estimated same slope is is 3.6 and StanError.96

The individual ones are 1.09, -4.9, 6.4, 3.4, -0.9, 10.8, 7.7 and from these 7 values the standard errors about 2.

Sometimes it will be more reasonable and the conclusion will probably remain unchanged.

The estimated same slope is is.25 and StanError.06

The individual ones are.25,.39,.11,.25,.2,.11,.11, and from these 7 values the standard errors about.04

2) When modelling individual events within the day (from Analysis 1.5 on) all observations within a day (up to 50) are assumed independent.

In such longitudinal data within the day correlations from observation to the next one are expected to be high.

This hampers analyses of individuals as well producing to optimistic standard errors and small p-values.

A simple way out is to construct a daily summary.

3) Adjusting for multiplicity is needed also when trying different models for heart rate, namely its dependency on different variables and in different subsets (1 month 2nd month etc)

With the current p-values not much will change in the conclusion, with new ones I do not know.

---

## [Author Response]

Reviewer #1:

This is an important longitudinal study that compare maturation rate of vocalization to movement and postures. Authors discoveries, if correct, are of general theoretical and practical interest. Of particular interest is the dynamic analysis of movements and calls coordination, showing reversal of patterns over development. Before this manuscript can be published, several concerns about the statistical approach and about conclusions should be addressed:A) Statistical approach:1) Using each call as independent statistic is problematic, and (to my knowledge) not an acceptable practice in acoustic analysis. At least for the most critical tests, the n must be the number of monkeys studied, namely 7.

We have revised the analyses to utilize sophisticated techniques - generalized linear mixed effect models - that control for multiple replicating variables (e.g., postnatal day and subject). Our rationale and description of the models are described in responses 2-4.

2) Authors should be more careful with their statistical modeling, and take a simple, direct approach whenever possible. In particular, when comparing time courses of calls and movement maturity indexes, author should simply test if those time courses are different. That is, the null hypothesis should be that those are the same. Here all we need is a pairwise comparison within each subject: is the zero crossing the same or not? Such pairwise comparison with n=7 using paired Wilcoxon, or even paired t or just binomial test would be (one can safely reject the null hypothesis (with p=1/2^7) and conclude that the maturation time course of vocalization is indeed faster.

We understand, appreciate and agree with the reviewer’s point. We wanted to make use of our complex developmental dataset that varies not just by behavioral category, but also by postnatal day and subject. We realize that our original testing regime was inappropriate because it treated each data point as an independent observation. To correct this error in the revised manuscript, we chose to use a sophisticated statistical technique – generalized linear mixed effect models (GLMM) – to make our comparisons. GLMMs are excellent for complex datasets, enabling the user to test for population-level fixed effects while controlling for random effect variables operating on the individual level. Below, we describe the GLMM model that we used and its outcome.

In the GLMM, we include behavior type (vocal vs. postural, and vocal vs. locomotor) as the fixed effect, with postnatal day and subject id as random effects. We used a ‘logistic’ regression function for this GLMM, as it is most appropriate for fitting sigmoid-shaped curves. We had to adapt our maturity index slightly in order to use a logistic GLMM. A logistic regression requires proportion data (i.e., falls within [0,1]). The maturity index in the original manuscript was scaled between -1 and 1. In the revised manuscript, we calculate the maturity index as the proportion of time spent engaged in adult-like behaviors. Thus, the data structure is still the same as before, just shifted to range between 0 and 1.

The outcome of the GLMM is in line with the outcome reported in the original manuscript. The vocal maturation occurs sooner than maturation of postural and locomotor behavior. Because the logistic GLMM analysis requires proportional data, the plots in Figures 2B,D and E have been rescaled. We also updated the significance indicators in plot E.

3) The way authors did the test is inappropriate in several ways. First, raising 3 distinct hypotheses (H1= more, H2=same, H3=less) is not necessary, it suffice to look for a significant difference from null. Second, and most critical – where do 124 degrees of freedom comes from? This looks like an obvious pseudo-replication, which resulted with illegitimate power of W = 3936 (!) and an erroneously huge significance (p < 0.0001). Such p values are rarely correct in biology, certainly not with the sample size of the current study.

We agree with the reviewer that our original analysis technique was fraught with pseudo-replication and therefore inappropriate. We apologize for this. To correct this problem in the revised manuscript, we used a generalized linear mixed effects model. Please see response 2 for a detailed description of our rationale, the model, and changes made to the manuscript.

4) Similarly, the analysis of movement vs. Wiener entropy should be done with each monkey as a statistic, and again, paired comparisons should be done within each animal.

Extending the rationale described above (response 2), we applied generalized linear models to analyses of movement vs. call duration and movement vs. Wiener entropy. We describe these two models in the Materials and methods. Briefly, we include both locomotor activity and postnatal day as fixed effects and infant identity as a random effect. We updated the Results and Figure 3B accordingly. The outcome of the new analyses has the same main outcomes as before -- locomotor activity is negatively associated with call duration and positively associated with Wiener entropy. These models also show that postnatal day is positively associated with call duration and negatively associated with Wiener entropy, which are findings that replicate previous studies.

5) The only place where I find it reasonable to use calls as statistics is in the dynamic analysis shown in Figures 4 and 5. In such a fine grain developmental analysis, and after the major effects where established, it makes sense to look at time courses of call movements interactions, considering each event as statistically independent.

We now use generalized linear mixed models (GLMM) for most analyses. Please see responses 2-4 for details.

B) Conclusions:This is an important, but still a correlational study – we don't know cause and effect. For example, the statement "infants must overcome the energetic costs of coordinating their vocalizations with body movement" is not fully supported by the data. For example, the marmoset is likely to vocalize while already agitated and this effect might be age related. Further, one might suspect that vocalization might be used to communicate high energy/agitation state.

In the revised manuscript, we use heart rate as a measure of arousal fluctuations rather than attempting to convert it to “energetic cost”. In addition, we removed causal statements when discussing the results. We agree with the reviewer that these statements are misleading given that our study design is observational. For example, we have re-written the Abstract statement mentioned above to “infants gradually improve coordination between vocalizations and body movement through a process that may be facilitated arousal level changes”. We also include a cautionary statement in the Discussion that “However, it is important to note that we are unable to test causal claims directly due to the observational design of this study”.

Reviewer #2:

[…] The paper includes several novelties: i) concurrent analyses of both the developmental timescale as well as the real-time (seconds) timescale to investigate energetic tradeoffs during vocalizations. ii) which allows to them study coupling for the first time. The authors show that this coupling exists at the second scale suggesting higher energy investment in calling behavior is required for producing mature calls. The data is well presented with clear figures.Major comments:1) The data is used to support the hypothesis that mature contact call production requires an energy investment that increases with age. In the light of energy expenditure during vocalizations this can be explained if the calls get louder. Increasing respiratory effort and thus lung pressure increases predominantly sound amplitude in laryngeally produced sounds. Fundamental frequency only changes slightly with pressure. Wiener entropy as a measure for noisiness is not regularly used outside of the birdsong field, but will increase most likely in chaotic regimes when pressures are very high.It would be rather straightforward for the authors to quantify the source levels of the vocalizations. The low-frequency cry is not directional but the wave-number can be used to coarsely estimate the directionality of the phee call. (ka = 2pi/λ * a (, where λ is wavelength (v/f) and a is emitter size radius – guessing 2 cm for a marmoset. with k=2pi/(340/7000) and a = 0.02) ka=2.5, which is above 1. Thus the phee call is directional, which makes it more complicated to compute source levels without microphone array. However luckily the microphone was located in the far-field 90 cm above the cage directing down, making vocalization's aimed in the horizontal plane directly comparable.

We appreciate these insights and we did not consider this approach to measuring sound amplitude as a more robust indicator of respiratory effort. We hope to use it in the future. The current revision, however, adheres to the requirement outlined in the decision letter that was to remove the entire energy expenditure line of argument/analyses. Thus, this comment, while very much appreciated, is no longer applicable to the revised manuscript. If you would still like us to do the analysis, nevertheless, then we’d be happy to give it try. Do let us know.

2) The authors list four empirical data papers in the last paragraph of the discussion that investigated human infant vocal-locomotor coordination. At least Berger et al. and Abney et al. present similar datasets of concurrent posture and vocalization (not at the second timescale). These papers should be acknowledged and placed in perspective up front in the Introduction.

Thank you for these references. We have added these papers to the Introduction in the paragraph where other human-focused studies are now discussed.

3) The locomotor behavior was basically scored as difference images. Did the authors observe any structural changes in locomotory patterns, such as certain accompanying displays (vocal postures) that matured over development?

We did not observe structural changes in the co-occurrence of specific postural-locomotor behaviors with vocalization across development. In an exploratory data analysis, we tested whether phees (mature contact call) were more likely to co-occur with mature forms of posture-locomotion than with immature forms. These analyses were focused on days in which monkeys produced phees and both immature and mature forms of posture (n = 23 sessions) or both immature and mature forms of locomotion (n = 27 sessions). The results suggested that Phee production did not depend on the maturity level of posture (p = 0.70) or locomotion (p = 0.84). Instead, it appeared that phee production in younger infants depended on whether infants were immobile.

Reviewer #3:

General comments:This manuscript describes a research effort, in which development of locomotor and vocal skills as well as, importantly, their coordination have been studied quantitatively and have been related to energetic aspects. This is a very interesting study that tries to fill a gap in our understanding of the ontogeny of motor system coordination and its energetic background. The presented data partly meet the goals set out in the Introduction (questions 1-3), but especially for questions 2 and 3 fall somewhat short and require clarification.1) The various sets of hypotheses listed in the results present sets that leave out potential alternatives. Specific suggestions are mentioned below.

To address several of these comments, we chose to drop the “energy expenditure” part of the manuscript. Instead, we focus on heart rate as an estimate of arousal dynamics.

2) The assessment of the energetic cost of call production is problematic at multiple levels, from the premises to the estimate and the interpretation. Specifically, the following general points should be considered:2.1) Several assumptions had to be made to convert heart rate to metabolic activity. We know from critical assessments of the technique that careful calibrations are needed to achieve reliable estimates. That means that the applied conversion factors may not be valid across development within an individual and certainly are problematic when applied across individuals. Whereas this is a general difficulty with the technique that cannot be solved in this study, it should be acknowledged and should lead to more careful conclusions.

We addressed this comment by using heart rate directly as an indicator of arousal rather than converting it to an estimate of energy expenditure.

2.2) It is not clear to me how baseline levels of heart rate have been determined for call production episodes with and without locomotor activity. Because heart rate changes are very dynamic, this is one of the critical methodological issues, which we face. Are observed changes due to actual changes or differences in baseline levels?

Thank you for this insight. In our study, we normalize heart rate by converting beats/min to percentiles. This normalization method preserves the heart rate changes within a session but allows the dynamic traces to be scaled around the same median (50^th^ percentile) between observation sessions. This is the same method used in a study of arousal dynamics and vocal production in adult marmosets (Borjon et al., 2016). Thus, we aren’t really comparing “baseline” levels of heart rate, we are comparing deviations of heart rate from session “medians”. In the revised manuscript, we describe this better in the Materials and methods and we refer to “median levels” instead of “baseline levels” in the Results.

2.3) Rapid changes in heart rate can be caused by motivational and other modulatory factors that have no direct link to energy expenditure of a behavior. This is a major shortcoming of using heart rate to assess the metabolic cost for short-duration behaviors.

We understand. In the revised manuscript, we used heart rate as an estimate of fluctuating arousal levels rather than of energy expenditure.

2.4) It is not clear to me how the cost of locomotion, which is expected to be several-fold greater than that for vocalization has been disentangled to conclude that production of phee calls imposes a high metabolic cost.

We appreciate the comment. Loud calls such as the marmoset contact call are likely to be very energetically costly for such a small animal (~300 grams in weight) in the same way that it is for other very small animals. Whether it is several fold less than locomotion depends on the duration and mode of locomotion. In any case, for the revised manuscript, the point is moot. We used heart rate as an estimate of fluctuating arousal levels rather than of energy expenditure.

3) The presentation of the results is interwoven by methods (somewhat necessitated by the fact that the method section is at the end) and lots of introductory material and posing of hypotheses. The actual results are then described very briefly, and statistical as well as descriptive presentation is quickly reduced to very derived spline curves. At least to me this generates the feeling that I do not know the results sufficiently to assess whether or not the later conclusions are supported sufficiently.

Thank you for pointing this out. We have removed the original energetic analysis from the revised manuscript, so part of this comment is no longer relevant. In the revised manuscript, we included more mainstream statistical analyses (generalized linear mixed models) so that readers can better understand the results. Some of these models are described above in responses 2-4. In addition to the models described in responses 2-4, we also use GLMMs to supplement the fine-grained dynamic analyses (described in Materials and methods). We believe these models provide more substance to the revised Results sections on the real-time and developmental dynamics between call production, locomotor activity, and arousal levels.

Specific comments:Abstract: Is the intended meaning that locomotion uses resources that would otherwise be available for call production? I cannot see that such an energetic bottleneck exists. If a constraint exists, it is more likely that locomotion and call production require either muscle efforts that are difficult to generate simultaneously or some other coordination-related trade-off. The absolute energy level is very unlikely to be the bottle neck here.

This is a great point that we did not consider. Indeed, we were focused on the idea that energy was the bottleneck. Sorry to repeat this over and over again but to be complete: In the revised manuscript, we used heart rate as an estimate of fluctuating arousal levels rather than of energy expenditure.

Introduction paragraph one: Of course, there are always energetic costs to vocal production. How much energy is required may vary, but as discussed above it is hardly so high that energy levels prohibit vocalization.

In the revised manuscript, we used heart rate as an estimate of fluctuating arousal levels rather than of energy expenditure.

Subsection “The vocal system matures before postural and locomotor systems” paragraph two: This description of hypotheses is almost "trivial" and thus does not need to be so elaborate.

Thank you for the suggestions. We simplified this section to describe briefly the “null” (overlapping) and “alternative” (non-overlapping) hypotheses. We deleted several redundant phrases (~50 words).

Subsection “Mature contact call production and locomotor activity become increasingly coordinated during development”: These hypotheses are based on the same notion that absolute energy levels at the time of vocalization could dictate whether or not calls can be produced. I do not see that such an energy shortage could exist. It is more likely (as alluded to above) that other factors dictate whether or not a call can be produced (e.g., can a particular respiratory muscle generate sufficient force if it also is engaged in locomotion, etc.). Another matter is whether or not an animal engages in a particular behavior because of general energetic condition. These conceptual issues are at the heart of this paper, and, in my opinion, the current reasoning is not realistic in regard to the physiological understanding of metabolic cost and movement.

Yes, we were wrong (or at least incomplete) in our thinking. We sincerely appreciate you pointing this out. Your insights not only affect the current manuscript (and the elimination of energetics from it) but also how we are thinking about on-going studies in our lab. In the revised manuscript, our interpretations are now focused more on how fluctuating levels of arousal predict motor output and coordination. We refrain from making strong causal statements in the revised manuscript.

Subsection “Mature contact call production and locomotor activity become increasingly coordinated during development”: This discussion would read very differently if the constraint were seen more like discussed above.

This discussion is now focused around using fluctuating arousal levels to predict motor output.

Final sentence of subsection “Mature contact call production and locomotor activity become increasingly coordinated during development”: This statement may have to be revised if calibrations at different ages were available.

Analyses on changes of “energy expenditure” across development have been removed from the revised analysis.

Materials and methods: It is not clear how baseline heart rate was determined. Because metabolic cost of locomotion is not restricted in time to the actual duration of movement, but elevated metabolism continues after the movement, the timing of call production relative to locomotion is very important. This issue is very critical, if the costs of locomotion and vocalization are to be disentangled. It is also not clear how z-scores were calculated to make data comparable over developmental time.

Please see response above for a discussion of what we meant by “baseline” and how we used heart rate percentiles to standardize arousal dynamics across sessions. We agree that dynamic changes in physiological condition are important. For this reason, we continue to include a version of Figure 4, which shows arousal fluctuations surrounding call production and locomotor activity (this analysis is new and described in the Results). Our data suggest that vocal-locomotor behaviors occur during arousal states that have shifted well before (> 10 s) the behaviors occur, and so averages should help to capture these more drastic changes. Given that the manuscript is no longer focused on trying to disentangle the energy costs of behaviors, the nuisances of timing should be less of a concern.

[Editors' note: responses to the first round of re-review follow.]

Reviewer #1:

We respectfully but wholly disagree with this reviewer’s assessment of our statistical approach but we appreciate the opportunity to clarify what we did. Moreover, we’d be happy to subject our analytical approach to the evaluation of anyone the editor views as a more expert statistician than us. Finally, it is worth noting that all our data will be available for anyone to analyze as they wish.

Authors addressed my comments and the manuscript is much improved but I am still worried about the statistics. Authors now use GLM throughout but they did not specify how dependent repeated observations are treated in the model.

We respectfully disagree with the reviewer that we did not specify how dependent repeated observations are treated in the model. As described in the Materials and methods, we include in the regression models the postnatal day and identities of the infants to control for the effect of those variables.

Judged by the p values, which are all lower than 0.0001, I suspect that p values are based on the dependent call events and movement events as the statistics, rather than the animal.

We understand the reviewer’s concern. As the reviewer knows, the p-value just tell us what is the probability that assuming the null hypothesis we would observe a value that is further a part in the distribution; it is arbitrary. Therefore, the reviewer's inference that the small (in his/her opinion) p-value is automatically a consequence of the "wrong" sample size is not correct. For example, if you have a sample of dependent variable that is [1,2,3] and a corresponding sample of a predictors that is [2,3,4] the correlation value is 1 and p-value is smaller than 0.00000001, although the sample size is only 3.

This is a problem. I will need to see the breakdown of within and between subject results, and how p values were computed to confirm, but looking at the raw data, I am almost sure that p values are wrong. Authors should keep in mind that as opposed to R, Matlab is not a statistical software, and it is very easy to get the model wrong. One should use Matlab statistical packages carefully, particularly when extracting p values from complex models, to make sure all assumptions are correct.

To address this concern, we carried out the regression analyses in R software (packages lme4 and LmerTest) instead of Matlab and updated the revised manuscript accordingly. The outcomes of the R models were compatible with the Matlab models—the p-values are correct. Perhaps this reviewer was concerned about the assumptions. To mitigate against this possibility, we repeated the regressions for individual subjects separately and found similar results. Additionally, we provide the data in Dryad so that readers can test the statistics on their own. Finally, we also cite the scholarship related to our statistical approach using logistic generalized linear mixed effect model in R:

Bates, D., Mächler, M., Bolker, B., and Walker, S. (2014). Fitting linear mixed-effects models using lme4. *arXivpreprintarXiv:1406.5823*.

Kuznetsova, A., Brockhoff, P. B., and Christensen, R. H. B. (2017). lmerTest package: tests in linear mixed effects models. *Journal of Statistical Software, 82*(13).

I still believe it would be much safer to shuffle statistics here: simply shuffle subjects across groups and extract direct p values. Simple is better than sophisticated.

Our analysis is standard in the developmental literature; it is not extraordinarily sophisticated. We disagree that reducing the longitudinal data to a single value by shuffling subjects is a better way. It might look simpler, but one would simply lose information by doing. From our perspective, it does not do justice to the data to reduce a time series to a mean-and-standard deviation.

Reviewer #3:

The manuscript has been improved following the suggestions. The metabolic data and associated argumentation have been taken out. In its place, the heart rate data were used to assess "arousal". "Arousal" is a vague concept and is not defined very well in this manuscript. Whereas I do not see the issues as critical as in the case of metabolism, I am not clear on whether or not the problem of disentangling metabolism related to locomotion from emotional influences has been solved any better here.

We apologize for not explicitly defining “arousal” in the manuscript. It is in fact a very clearly defined concept even to the degree that investigators can study the genetics of arousal (see the work of Donald Pfaff of Rockefeller University). We have used “arousal” as a focus in three of our recent manuscripts without incident (Zhang and Ghazanfar, 2016; Borjon et al., 2016; Liao et al., 2018 PNAS).

In the revised Introduction, we define “arousal” in the following manner: “An animal would be said to exhibit a high arousal state if it is more alert to sensory stimuli, more motorically active and more reactive (Pfaff, 2006). The role of arousal is essentially a means of allocating metabolic energy (i.e., preparing the body for action).”

It is worth noting that “arousal” is separate from “emotions” or “affect”. Both positive and negative affects could be associated with the same level of arousal.

Perhaps I still do not understand it correctly, but the issue of baseline data has in my opinion also not been resolved satisfactorily. Namely, the interpretation of whether or not and how much heart rate increased depends on the period of baseline measurement.

With apologies, we see that our use of the word “baseline” in the revised manuscript is still quite misleading. What we meant in those areas of the manuscript is that the observed real-time and developmental fluctuations in heart rate are outside the 95% threshold of bootstrapped significance tests (i.e., shuffled permutations of the data). In the revised manuscript, we have taken out the word “baseline” completely and now refer specifically to how observed data changes relative to the 95% threshold of the bootstrap significance tests.

I am not sure I understand what a session is (letter) for which the normalized data are taken. Is it the period before and after as outlined in the manuscript?

“Sessions” are 10-minute observations (taken every ~1-3 days) of infants. All behavioral and physiological data presented in this manuscript were collected during these observation sessions and not from before and/or after the observation sessions. In the revised manuscript, we are more careful to refer to sessions specifically as “observation sessions” so that the intended meaning is clearer.

Certainly, different locomotor activity during these periods will make comparisons very difficult. Because locomotor behavior increases with maturity level, so will the associated heart rates. To what degree it is a change in arousal, can therefore only be said with confidence if the two influences on heart rate can be disentangled.

Thank you for this comment. We believe we created some confusion regarding how we re-scaled the heart rate data so that fluctuations can be compared within and across observation sessions. We describe this process in more detail in the Materials and methods section.

Locomotor activity also goes through a re-scaling process within observation sessions, which we describe in more detail in the Materials and methods. It is important to stress here that we are not attempting to compare absolute levels of heart rate and locomotion but rather relative levels of heart rate (percentiles ranging from 0 to 100) and locomotion (smoothed binary ranging from 0 to 1) fluctuations (Appendix 1). In other words, if the heart rate percentile during a contact call is a 100 and the locomotor activity is a 1, that would mean that an infant was mobile with a high heart rate relative to the all data from the same 10-minute observation session.

Another concern this reviewer may have is that any associations between heart rate and locomotor activity during contact calls are a mere by-product of age. To mitigate this possibility, we tested associations between heart rate and locomotion (during phee calls) for one- and two-month old infants separately. We found a positive association for both age groups and report these findings in the revised manuscript:

“Moreover, this positive association characterized infants that were both one month old (1-30 days; n = 1,705 calls, β ± SE = 14.84 ± 4.62, t = 3.21, p = 0.0013) and two months old (31-61 days; n = 2,261 calls, β ± SE = 7.19 ± 3.51, t = 2.05, p = 0.0407). This means that a positive association between locomotor activity and heart rate percentiles was not simply a by-product of age.”

[Editors' note: responses to the second round of re-review follow.]

Reviewer #4:

Major concerns:1) Authors use LMM which is fine. They say they have infants as random effect and that is fine too, but I wonder how could authors get with 7 monkeys p<0.0001? I looked at the tables with per-animal results. The variability is high in the coefficients per monkey, sometimes with results of opposite directions. Treating them as a sample does not yield the precision near to that claimed.

We understand this reviewer’s concern and we made effort to make the analysis and code more transparent and reproducible. We think that making the data, code, and the model assumptions clear should solve this issue. The reader can now easily check our statistical results and be convinced (or not) by our evidence. The complete answer is in the response to the next comment.

Because this issue with the interpretation of the p-value came out in the previous reviewing round too, and we failed to explain it adequately (partly because we failed to provide the code to the reviewer), we wanted to clarify some points that we think are relevant. Treating each monkey regression coefficient as a sample makes the analysis underpowered. The reason is because the coefficient is not a “single data point”, but the result of a regression line fitted to several data points. As an extreme example, if we have two regression lines and wanted to test whether the coefficients were different or not, we would not say that the sample size is one in each group, because that would preclude any reasonable statistical inference. Obviously, our situation is more complex as we would like to make a generalization to the population. That is why we considered a mixed effect model to increase the statistical power (type II error) of our models. Maybe at more fundamental level, the p-value is not a measure of precision nor of the strength for evidence of the claims. It is simply the probability that we observe a value as extreme as in the data given that the null hypothesis of no effect is correct. The value will depend on the distribution of the alternative hypothesis that is unknown.

2) I cannot exclude the possibility that authors may be right, BUT when submitting such an analysis reproducible computing is a must. The statistical model itself should be specified and the entire code made available. I think that giving this information is minimal requirement, but making the code and the data available at this stage so that reproducibility could be checked is already a requirement made by leading journals and here it is crucial.

We agree with the reviewer that it is crucial that the analyses are more transparent and reproducible. We have addressed this concern in three ways. First, we describe the linear regression models in greater detail in the Data Analysis section. We explicitly state every component of the models (dependent variables, fixed effects, and random effects) and include the model equations used to generate the results in the R script. Second, in Appendix 1, we provide detailed R software output from the population-level regression models (formulas, random effect variance, fixed effect and intercept estimates, standard errors, t values and p values), as well as the output from models run on individual monkeys. Having this appendix will allow interested readers to examine several aspects of the regression models without having to download the datasets and R/Matlab scripts. Third, with the current submission we provide via dryad all data and code needed to run the regression models (R script and.csv data files) and generate the figures (Matlab script and.mat data files).

3) In particular in this case, it is important to know whether the model considered sessions as nested within monkeys or not. It will have great impact, and you cannot tell it from the information authors provided.

We agree with the reviewer that the treatment of random effects as nested or not could have great impact. To avoid any doubt, we followed the recommendation to keep the mixed effect structure maximal (Barr et al., 2013). This means in particular that, whenever possible, we have the variables as both fixed and random effect, keeping the nesting. We re-formulated the majority of regression models to include postnatal day (i.e., sessions) nested within infant subjects (i.e., monkeys) as the random effect. We now include more detailed descriptions of the regression models and their formulas so that it is clear that the nested random effects are used (see response 2). The new regression models had a subtle influence on the statistical outcomes (updated estimates, t-values, and p-values are presented). However, these changes were not large enough to sway the statistical significance at the p < 0.05 level or our interpretation of the models.

Barr DJ, Levy R, Scheepers C, Tilye HJ (2013) Random effects structure for confirmatory hypothesis testing: Keep it maximal. *J Mem Lang* 68: 255-278. DOI: https://doi.org/10.1016/j.jml.2012.11.001

4) In the correlation analysis authors use all observations as if they are independent and that is wrong, but since it is used only for creation of scale it does not matter much.

We have updated these analyses to use linear mixed models, which mirrors the analysis approach taken in the rest of the manuscript. We describe these regression models in the Data Analysis section, and report the findings in Table 1. The significance outcomes and interpretation did not change by switching over to LMMs from spearman correlations.

5) Authors also present too many figures only with no data, only summaries: all of 4 and 5B.

We adapted Figures 3, 4, and 5 to include data instead of only summaries. In Figures 3B and Figure 5B, we include raw data points in addition to regression slopes and confidence intervals. In Figures 3C, 3D and all of 4, we include additional subplots to illustrate raw data for each individual.

6) eLife requires treatment of multiplicity issues; none of that was done.

In the revised manuscript, we applied a Bonferroni-Holmes correction to adjust p-values in instances where multiple models were used to answer a similar question. Specifically, we applied this correction when examining proportions of time engaged in various behaviors across development (Table 1), associations between locomotor activity (during contact calls) with multiple acoustic parameters, associations between locomotor activity (during contact calls) for different call types, and associations between ANS activity (during contact calls) for different call types. Applying this correction was useful for making p-values more conservative. However, the changes made to p-values were not enough to alter outcomes and interpretation.

[Editors' note: responses to the third round of re-review follow.]

Reviewer #4:

Reviewer notes on "Vocal and Locomotor coordination…" by Gustison et al.I thank the authors for their response to my previous questions. Their effort to make their analysis transparent and reproducible are evident in this version. I hope this version can serve as a model for reproducible research for others.These compliments do not mean that there are no problems with their analyses,I shall list three of them:1) The model they used allows only random intercepts, and not random slopes.hence the variability from one monkey to the other is not reflected in the calculations of the slope's standard deviation and p-value, which are far too small. In order to enhance replicability of their results, when the experiment will be conducted on another set of monkeys the random slopes inference is important (Then the models should be ~Day + Day/Subject)The analyses for individuals is appropriate so it lets assess how serious this flaw is.Sometimes it is very inappropriate: HeartRatePercentileMean~CallType_Cry0Phee1The estimated same slope is is 3.6 and StanError.96The individual ones are 1.09, -4.9, 6.4, 3.4, -0.9, 10.8, 7.7 and from these 7 values the standard errors about 2.Sometimes it will be more reasonable and the conclusion will probably remain unchanged.The estimated same slope is is.25 and StanError.06The individual ones are.25,.39,.11,.25,.2,.11,.11, and from these 7 values the standard errors about.04

Following the reviewer’s recommendation, we changed the generalized and linear mixed model formulas to include random slopes (described in detail in Material and Methods – Data Analysis). These changes to the model formulas resulted in more conservative estimates, standard errors, t-values, and p-values. The main conclusions of our study did not change.

a) The vocal behavior develops faster than posture/locomotion behaviors.

b) Locomotor activity is reduced during mature calls; vocal-locomotor coordination improves across development.

c) Mature calls are produced at elevated arousal; arousal levels are higher during vocal-locomotor coordination.

We are now confident that our statistical output is an accurate representation of the data structure and is fully reproducible.

2) When modelling individual events within the day (from Analysis 1.5 on) all observations within a day (up to 50) are assumed independent.In such longitudinal data within the day correlations from observation to the next one are expected to be high.This hampers analyses of individuals as well producing to optimistic standard errors and small p-values.A simple way out is to construct a daily summary.

Following the reviewer’s advice, we constructed daily summaries and included them in the Appendix. These summary tables include per Day results where appropriate (i.e., for models with Day as a random effect term). The previous version of the manuscript only emphasized Subjects as a random effect term, whereas both Day and Subject are emphasized in the revised manuscript.

3) Adjusting for multiplicity is needed also when trying different models for heart rate, namely its dependency on different variables and in different subsets (1 month 2nd month etc)With the current p-values not much will change in the conclusion, with new ones I do not know.

Following the reviewer’s recommendation, we’ve adjusted p-values to account for multiplicity issues in models involving heart rate. We still find an association between locomotor activity and arousal level during first month of development.